# Data Diversity as Implicit Regularization: How Does Diversity Shape the Weight Space of Deep Neural Networks?

## Abstract

Data augmentation that introduces diversity into the input data has long been used in training deep learning models. It has demonstrated benefits in improving robustness and generalization, practically aligning well with other regularization strategies such as dropout and weight decay. However, the underlying mechanism of how diverse training data contributes to model improvements remains unknown. In this paper, we investigate the impact of data diversity on the weight space of deep neural networks using Random Matrix Theory. Through spectral analysis and comparing models trained with data augmentation, dropout, and weight decay, we reveal that increasing data diversity alters the weight spectral distribution similarly to other regularization techniques, while displaying a pattern more closely aligned with dropout than with weight decay. Building on these insights, we propose a metric to explain and compare the benefits of diversity introduced by traditional data augmentations and those achieved through synthetic data.

## 1 Introduction

Data augmentation, achieved through purposeful manipulations of training data to increase diversity or variation, is a widely adopted practice in deep learning and has been shown to significantly improve model generalization (Bishop, 1995). Empirical evidence (Thulasidasan et al., 2019; Rebuffi et al., 2021) suggests that increasing training data diversity yields a similar reduction in overfitting as traditional regularization methods, such as *dropout* (Srivastava et al., 2014) and *weight decay* (Krogh & Hertz, 1991). Data augmentation techniques, including adding noise, rotation, cropping, random erasing, and color jittering for vision tasks (Shorten & Khoshgoftaar, 2019), and synonym replacement, word insertion, and deletion for language tasks (Wei & Zou, 2019), are employed to modify input data in deep neural networks (DNNs). In contrast, dropout reduces model complexity by randomly deactivating neurons during training, while weight decay regularizes the model by adding a penalty to the loss function based on the magnitude of the weights (Zhang et al., 2018; Andriushchenko et al., 2023).

However, despite the existing evidence of their effectiveness, the underlying mechanism for data augmentation remains less discussed in the literature. Given that these regularization methods all seek to improve generalization, albeit through different approaches, a question naturally arises: *Does data augmentation have similar effects on the neural network's weight distribution as dropout or weight decay?*

In addition, a recent trend in AI and deep learning is to use generative models, such as GANs and LLMs, to generate synthetic data to augment the original real dataset. The effectiveness of this approach in improving model performance has been demonstrated across various domains (He et al., 2022; Azizi et al., 2023). This motivates us to investigate the following question: *Is there a difference in the diversity introduced by synthetic data augmentation compared to traditional techniques?*

To answer these questions, we employ Random Matrix Theory (RMT), a statistical framework for analyzing the structure of large matrices, to measure the changes in weight matrices across different regularization approaches. Martin & Mahoney (2019) indicates that the weight matrix $W$ from a specific layer in neural networks, either through extensive training on labeled data or fine-tuning on a downstream task, can be approximated as the sum of a random noise matrix $W_{\text{rand}}$ and a signal matrix $W_{\text{signal}}$, such as

$$W \simeq W_{rand} + W_{signal}.$$

Matrix $W_{\text{signal}}$ provides meaningful structural patterns that represent learned features, which are distinct from the random noise component $W_{\text{rand}}$. Therefore, examining $W_{\text{signal}}$ could provide insights into the effects of data diversity on the learned weight matrices (Pennington & Worah, 2017; Sagun et al., 2017), facilitating comparisons of robustness among regularization approaches. RMT provides tools for analyzing the spectral properties of weight matrices. If data augmentation results in spectral characteristics similar to those observed with dropout or weight decay, it suggests that augmentation offers a comparable implicit regularization effect.

In this study, we hypothesize that the diversity of the input $X$, measured by the Vendi Score (VS) (Dan Friedman & Dieng, 2023), influences the spectral distribution of the weight matrix $W$, which in turn affects model complexity and generalization. To test this hypothesis, we conduct extensive experiments across both vision and language tasks, involving both fine-tuning pre-trained models and training models from scratch. Our empirical results show that increasing data diversity affects the spectral distribution of weight matrices in a way similar to regularization. This implicit regularization effect aligns more closely with the weight matrix behavior observed under dropout than with weight decay during training. Additionally, we develop a modified version of the Vendi Score to better highlight the strengths of different data augmentation strategies. This metric enables us to investigate how the diversity introduced by traditional and synthetic data augmentation impacts the model performance.

Our contributions are summarized as follows:

- We characterize the changes in the weight space of DNNs resulting from varying levels of data diversity;

- We provide theoretical arguments and empirical insights for explaining why diversifying training data improves model generalization;

- We design a metric that quantifies the effectiveness of diversity introduced by both traditional and synthetic data augmentation methods, shedding light on how each of them contributes to model performance improvement in distinct ways.

Our study provides deeper insights into the roles of data augmentation and data diversification, paving the way for their integration into modern deep learning workflows.

## 2 Preliminary

**Weight Decay.** Weight decay (Krogh & Hertz, 1991; Ishii & Sato, 2018) is a widely used technique to prevent deep learning model overfitting. It works by adding a small penalty $\alpha$, which is proportional to the size of the weight parameters, to the loss function during training. This encourages the model to keep weights smaller. During training, the weights are updated not only based on the prediction error-induced losses but also on this penalty, which pushes the model to find simpler solutions. Let $X$ represent the input data matrix, $X \in \mathbb{R}^{n \times d}$, with $n$ training samples and $d$ features and $y$ be the corresponding target vector, $y \in \mathbb{R}^n$, while $w \in \mathbb{R}^d$ is the model's learnable parameter matrix. By this technique, the objective function becomes

$$\underset{w}{\text{minimize}} \; \|y - Xw\|^2 + \alpha \|w\|^2 \tag{1}$$

**Dropout.** Dropout helps minimize overfitting during training by randomly disabling neurons in fully connected layers based on a given probability $p$. This randomness is controlled by a Bernoulli distribution, where neurons are either kept or deactivated during each training iteration. However, during inference, all neurons are active. To ensure consistency between training and inference, the output is scaled by $1 - p$ during training. Srivastava et al. (2014) demonstrates that, in expectation, the objective function of dropout

applied in linear regression is equivalent to $L_2$ regularization with a particular form, such as

$$\underset{w}{\text{minimize}} \; \|y - pXw\|^2 + p(1-p)\|\Gamma w\|^2 \tag{2}$$

where $\Gamma = \left(\text{diag}(X^\top X)\right)^{1/2}$. Therefore, dropout is expected to reduce the model parameters in a manner similar to weight decay.

**Data Augmentation.** Data augmentation (Zhang, 2017; Shorten & Khoshgoftaar, 2019) is a strategy to artificially increase the diversity of training datasets without requiring the acquisition of new data. This technique involves applying a range of transformations to existing samples, such as flipping, cropping, adjusting brightness, adding noise, and mixup. By introducing these variations, data augmentation enables models to learn more robustly from a wider array of instances, improving generalization. Synthetic data offers an alternative approach to expanding training datasets by generating realistic and diverse samples through generative models like GANs, VAEs, or diffusion models. Unlike traditional augmentation, this method synthesizes entirely new data points, enabling broader coverage of the data distribution (Tian et al., 2023).

**Spectral Analysis of Weight Matrices.** The Empirical Spectral Distribution (ESD) describes the distribution of eigenvalues of a matrix. A power law (PL) is a type of statistical distribution in which the probability of an event decreases polynomially with its size. In this study, we observe that the ESD of weight matrices can be well modeled by the PL distribution, particularly in characterizing its tail behavior (Eq. 3). Specifically, the ESD will be separated into bulk (left) and tail (right) parts, representing noise and signal, respectively, as demonstrated in Appendix D.1.

## 3 Methodology

In this section, we provide an overview of methods for measuring data diversity, the fundamentals of random matrix theory, and intuitive insights into the impact of data diversity.

### 3.1 Diversity Measure

We use Vendi Score (VS) (Dan Friedman & Dieng, 2023) to assess the data diversity introduced by data augmentations. This score quantifies the similarities among the data in a dataset, derived by using a set of samples along with a pairwise similarity function. Specifically, VS is defined as the exponential of the Shannon entropy computed from eigenvalues of the scaled matrix $X^\top X$:

$$VS = \exp\left(-\sum_{i=1}^{n} \lambda_i \log \lambda_i\right)$$

where $\lambda_i$ are the eigenvalues of scaled $X^\top X$. The higher the Vendi Score, the more diverse the dataset is. VS also provides customization for different similarity functions, such as embedding transformation for images and texts. We will correlate VS with the spectral changes in weight matrices to establish their relationship. In addition, we will introduce the Weighted Vendi Score ($\widetilde{VS}$) to better capture and explain the different diversities induced by traditional augmentations versus those induced by synthetic data.

### 3.2 Random Matrix Theory

We are concerned with the weight matrices of neural network layers. Given a weight matrix $W_l \in \mathbb{R}^{n \times m}$, we can construct its correlation matrix $\Sigma_i \in \mathbb{R}^{m \times m}$. Omitting the subscripts for simplicity, $\Sigma = \frac{1}{n}W^\top W$. The eigenvalues of $\Sigma_i$ are given by $\{\lambda_j\}_{j=1}^{M}$.

Let $p(\lambda)$ denote the Empirical Spectral Distribution (ESD) of a correlation matrix. According to RMT, the ESD of a Gaussian-noise random weight matrix follows the Marchenko-Pastur (MP) distribution when the dimension of a random matrix grows large. The MP distribution often exhibits a bulky shape. However, the empirical study of Martin & Mahoney (2021) showed that, for a well-trained DNN model, the ESD of

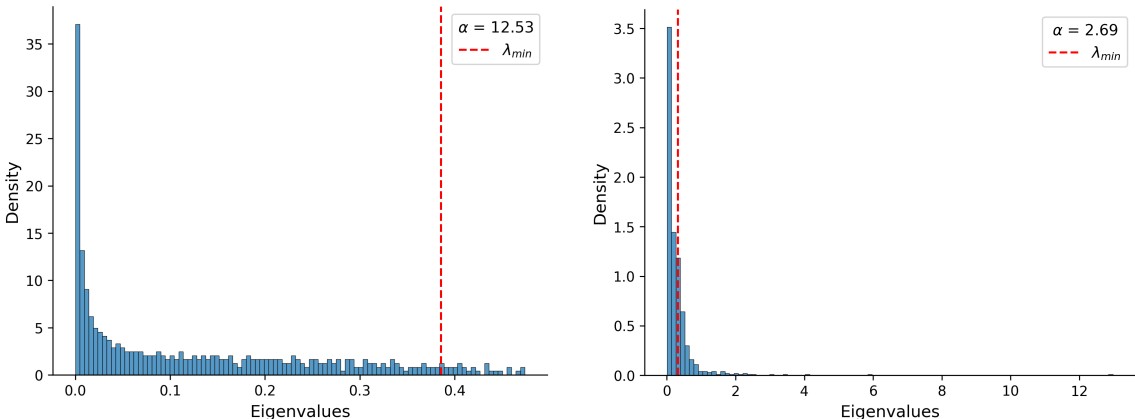

Figure 1: Plots of ESDs from feedforward layers of a fine-tuned CLIP model on CIFAR-100. The bulk and tail parts are separated by the selected $\lambda_{min}$. Clearly, $\alpha$ guides the shape of the distribution. A large $\alpha$ (**Left** panel) leads to a distribution of eigenvalues to be more evenly spread, while a small $\alpha$ (**Right** panel) leads to a more "heavy-tailed" distribution of the eigenvalue spectrum. Note that the scales of the horizontal axes of these two plots are different.

its weight correlation matrix is heavy-tailed and does not follow the Gaussian noise-based MP distribution. They called it the Heavy-Tailed Self-Regularization theory. Specifically, a power law (PL) distribution can be used to fit the heavy tail, which is given by

$$p(\lambda) \propto \lambda^{-\alpha}, \quad \lambda_{\min} \leq \lambda \leq \lambda_{\max} \tag{3}$$

Here, $\lambda$ takes values in the interval $[\lambda_{\min}, \lambda_{\max}]$, and $\lambda_{\max}$ is chosen to be the maximum eigenvalue of the empirical correlation matrix, while $\lambda_{\min}$ is selected to obtain a better power-law fitting, which is generally not equal to the minimum eigenvalue.

The weight matrix $W$ can be expressed as $W \simeq W_{rand} + W_{signal}$, where $W_{rand}$ represents "noise" and $W_{signal}$ represents "signal". If the weight matrix $W$ is dominated by random noise, the ESD of $W$ follows the MP distribution. But when structured signals are present, the ESD exhibits heavy-tailed behavior, which can be effectively characterized by a PL distribution. The weight matrix of a well-trained model is supposed to capture the intrinsic correlations between data features. Its eigenvalue spectrum typically exhibits a heavy-tailed pattern, where a small number of large eigenvalues deviate significantly from the bulk distribution. These outliers reflect the structured components of the weight matrix, $W_{signal}$, which encode meaningful signals or learned features. The remaining eigenvalues form the random bulk, $W_{rand}$, corresponding to noise or less significant directions (see Figure 1). The exponent $\alpha$ of a fitted PL distribution characterizes the tail behavior of the ESD. It has been shown to be closely related to the model's generalization capability (Martin et al., 2021; Yang et al., 2023b). A smaller $\alpha$ represents a more "heavy-tailed" distribution of ESD. This means that the tail of the eigenvalue spectrum contains more large eigenvalues that dominate and exert a stronger influence on the model's parameter space.

In this work, we explore the role of data diversity as an implicit regularizer by analyzing the spectral properties of the weight matrix $W$ from both the **scale** and **shape** metric perspectives of eigenvalue distribution. *Scale* metrics measure the magnitude of eigenvalues or the overall size of the weight matrix, while *shape* metrics describe the distribution shape and variability of the eigenvalues. Our scale metrics include the Frobenius Norm and Spectral Norm (Martin & Mahoney, 2019), as defined below.

$$\text{Frobenius Norm:} \quad \|W\|_F^2 = \|\Sigma\|_F = \sum_{i=1}^{M} \lambda_i$$

$$\text{Spectral Norm:} \quad \|W\|_\infty^2 = \|\Sigma\|_\infty = \lambda_{\max}$$

Additionally, our shape metrics include the power law exponent $\alpha$, as defined in Eq.(3), and the matrix entropy, as described below.

$$\text{Matrix Entropy:} \quad H(W) = -\sum_i \lambda_i \log(\lambda_i)$$

Table 1 in Appendix D.2 summarizes how the changes in these metrics may reflect the alterations in the structure of weight matrices and their associated spectral properties. The metrics mentioned above are calculated by WeightWatcher(Martin et al., 2021).

### 3.3 Spectral Analysis of Regularization Effects

Consider the classical problem in least squares regression: $X \in \mathbb{R}^{n \times d}$, with $n$ training samples and $d$ features and $y$ is the corresponding target vector, $y \in \mathbb{R}^n$; and $w \in \mathbb{R}^d$ represents the model's learnable parameters. For ordinary least squares, the optimal weight vector is

$$w = (X^\top X)^{-1} X^\top y$$

Writing $X^\top X = V \Lambda V^\top$ (Note that $X^\top X$ is positive semidefinite and $V \Lambda V^\top$ is its eigenvalue decomposition. The columns of the orthogonal matrix $V$ are the eigenvectors $v_i$, and $\Lambda = \mathrm{diag}(\lambda_i)$ is the diagonal matrix of corresponding eigenvalues). Then,

$$w = V \Lambda^{-1} V^\top X^\top y$$

The squared norm of $w$ can be expressed as

$$w^\top w = y^\top X V \Lambda^{-2} V^\top X^\top y = \sum_{i=1}^n \frac{q_i^2}{\lambda_i^2} \tag{4}$$

where $q_i = v_i^\top X^\top y$ is the projection of $X^\top y$ onto the eigenvector $v_i$. Eq.(4) highlights that directions with large data variance (large $\lambda_i$) contribute proportionally (via $\lambda_i^{-1}$) to the weight norm if the corresponding projections $q_i$ are non-negligible. When the target projection vector $X^\top y$ is approximately isotropic in the eigenbasis of $X^\top X$ (i.e., the energy of the target is uniformly spread across all eigen-directions), then the projection coefficients $q_i$ can be treated as uncorrelated with the eigenvalues $\lambda_i$. Under this simplifying assumption, we can write:

$$\mathbb{E}[w^\top w] \propto \sum_{i=1}^n \frac{1}{\lambda_i^2}$$

Regularization techniques modify these eigenvalues to control the scale of $w$: weight decay (ridge regression) replaces $\Lambda^{-1}$ with $(\Lambda + \alpha I)^{-1}$, yielding a modified weight matrix $w_{wd} = V(\Lambda + \alpha I)^{-1} V^\top X^\top y$. Similarly, dropout can be interpreted as adding a regularization effect (see Eq.(2)) so that $w_{dp} = V(\Lambda + (1 - p) \Gamma^2 I)^{-1} V^\top X^\top y$. Both regularizations act by editing the spectrum, effectively shrinking the weight norm.

By contrast, increasing data variance (e.g., via data augmentation) could indirectly inflate the eigenvalues of the correlation matrix. In this context, we focus on augmentation methods that do not introduce new samples into the training set. Data augmentation, such as Gaussian noise, brightness, blur, or small rotation, can be approximated as adding a perturbation matrix $E$, so augmented data $\tilde{X} \approx X + E, E \in \mathbb{R}^{n \times d}$. Each column of $E$ encodes the transformation applied to the corresponding feature dimension. The sum of the eigenvalues of the augmented data is then

$$\sum_i \tilde{\lambda}_i = \mathrm{tr}\big((X + E)^\top (X + E)\big) = \mathrm{tr}(X^\top X) + 2 \, \mathrm{tr}(X^\top E) + \mathrm{tr}(E^\top E) \tag{5}$$

If the augmentation noise $E$ is zero-mean and uncorrelated (or independent) with the data, then $\mathbb{E}[X^\top E] = 0$ and $\mathbb{E}[E^\top E] > 0$. Therefore, in expectation,

$$\mathbb{E}\left[\sum_i \tilde{\lambda}_i\right] = \sum_i \lambda_i + \mathbb{E}\big[\mathrm{tr}(E^\top E)\big] > \sum_i \lambda_i$$

This implies that additive, zero-mean perturbations increase the total variance of the data. When augmentations are not independent of the data (e.g., blur or small rotations), the cross-term $\mathrm{tr}(X^\top E)$ in Eq. 5 can be positive or negative. A well-aligned augmentation that perturbs the data along existing feature directions tends to satisfy $\mathrm{tr}(X^\top E) > 0$, further increasing the total variance, and $\sum_i \tilde{\lambda}_i > \sum_i \lambda_i$ holds. In

contrast, misaligned augmentations can distort the data manifold and cause undesirable distribution shifts. For non-additive transformations such as cropping, the augmentation can be represented as a linear mapping $\widetilde{X} \approx XP$, where $P$ encodes the transformation. The resulting total variance becomes

$$\sum_i \tilde{\lambda}_i = \mathrm{tr}\big((XP)^\top(XP)\big) = \mathrm{tr}(P^\top X^\top XP) = \mathrm{tr}(X^\top XPP^\top)$$

Here, $PP^\top$ determines how the transformation reshapes or amplifies variance across feature dimensions. If $PP^\top \preceq I$, as in cropping, the total variance decreases; Conversely, if $PP^\top \succeq I$, as in color jitter, it amplifies the variance. In this case, the eigenvalue sum is not guaranteed to increase. For example, random cropping typically reduces the total variance by discarding informative regions of the image. We empirically validated this theoretical observation using various transformations provided by *torchvision* on the CIFAR-10 dataset (Table 9). Thus, data augmentation that injects new variations into the data inflates the eigenvalue spectrum $\{\lambda_i\}$, which in turn reduces the weight norm.

Owing to the inherent complexity of deep neural networks, establishing direct analytical connections between the input data and the weight space throughout all layers remains a significant challenge. A study by Pennington & Worah (2017) states that in some cases (when expectation of the derivative of activation $f$, $(\mathbb{E}[f'(z)])^2 = 0$ ), the limiting eigenvalue distribution of the covariance matrix at layer $l$, $(Y^l)^\top Y^l$, does not distort that of the input covariance $X^\top X$ even after many layers [1]. Equivalently, the nonlinearities under this assumption behave in a way

$$\lim_{n\to\infty} \mathrm{spectrum}\left(\left(Y^{(l)}\right)^\top Y^{(l)}\right) = \lim_{n\to\infty} \mathrm{spectrum}\left(X^\top X\right)$$

It implies that the eigenvalues of the correlation matrix can be approximately preserved through the non-linear layer. The nonlinear RMT framework of Pennington & Worah (2017) is developed under idealized assumptions, and it is validated primarily in controlled or simulated settings. Nevertheless, it provides a theoretical intuition on how the correlation structures of weight matrices may propagate through neural network layers. Motivated by this theoretical insight, we will empirically demonstrate the regularization effects of data augmentation techniques on DNN learning, showing that a deep network trained on augmented data $\widetilde{X}$ exhibits a *built-in spectral regularization* effect. While our experiments do not enforce the exact assumptions of their analysis, we observe similar spectral behaviors in large-scale models (CLIP, ResNet, BERT) trained on real datasets, thereby extending their theoretical intuition into practical deep learning settings.

## 4 Experiment & Analysis

### 4.1 Experiment Setup

To evaluate the effects of different regularization methods, we use the relative changes in spectral measures compared to the baseline, where no regularization is applied. Let $M$ be one of *scale* or *shape* metrics, $M \in \{\|W\|_F^2, \|W\|_\infty^2, \alpha, H(W)\}$, and we compute its difference between pre- and post-training, i.e.,

$$\Delta M = M_{post} - M_{pre}$$

$M_{post}$ represents the metrics in the spectral domain after fine-tuning DNNs or training DNNs from scratch, and $M_{pre}$ represents these metrics before fine-tuning or initial training, with or without regularization or data augmentation. For example, $\Delta\alpha = \alpha_{post} - \alpha_{pre}$ quantifies the pre-to-post change in the eigenvalue distribution in the tail of the empirical spectral density. We track $\Delta M$ through layers and show their average values respectively. To draw our conclusions, we perform a comparative analysis of dropout, weight decay, and data augmentation. Treating the distribution of $\Delta M$ across layers for each category as time series data, we apply the adjusted ANOVA (F-test) and autocorrelation-aware pairwise t-tests to assess their statistical similarities between categories.

---

[1]See Pennington & Worah (2017), Section 3.1–3.2, for the derivation of the Marchenko–Pastur limit when $\zeta = 0$.

The dropout rate in the baseline model is zero, and we subsequently deploy the dropout rates of 0.1, 0.3, 0.5, and 0.7 to conduct a comparison study. The weight decay values are varied over six levels — 1e-5, 5e-5, 1e-4, 5e-4, 1e-3, and 5e-3 — to investigate how they affect the weight matrices. We also report their combinations with data augmentations, as shown in Figure 15 in Appendix I.

### 4.2 Regularization Effects of Data Augmentation

**Vision Tasks.** Our experiments consider two learning strategies under different model architectures: transfer learning and training from scratch. For the pretrained model, we employ the state-of-the-art text-to-image model CLIP (Radford et al., 2021) given its powerful capability of performing image classification tasks. We fine-tune CLIP (VIT-B32 and ResNet50 as its backbones) on CIFAR-10, CIFAR-100 (Krizhevsky et al., 2009), Stanford Cars (Krause et al., 2013), DomainNet (Peng et al., 2019). During the fine-tuning with 5 epochs, we freeze the text encoder; thus, only the weights in the image encoder and the last classification layers are updated. In addition, we train a ResNet-18 from random initialization on the CIFAR-10 dataset in 10 epochs (see Figure 10 in Appendix E).

To alter training data diversity, we implement ten different types of data augmentation: First, four automatic image augmentation techniques are applied: *AutoAugment* (Cubuk et al., 2018), *RandAugment* (Cubuk et al., 2020), *AugMix* (Hendrycks et al., 2019), and *TrivialAugment*(Müller & Hutter, 2021). Then we add *Gaussian noise* into the original images with a ratio of 0.1, 0.3, and 0.5. Finally, we created three advanced augmented datasets by combining *AutoAugment* with other data augmentation methods. They are *auto-v1*: AutoAugment + Gaussian noise with 0.5 ratio; *auto-v2*: AutoAugment + Gaussian noise with 0.5 ratio + RandomHorizontalFlip + RandomVerticalFlip; *auto-v3*: AutoAugment + Gaussian noise with 0.5 ratio + ElasticTransform. We monitor their VS values to confirm that these data augmentation methods provide certain degrees of data diversification, as shown in Figure 2.

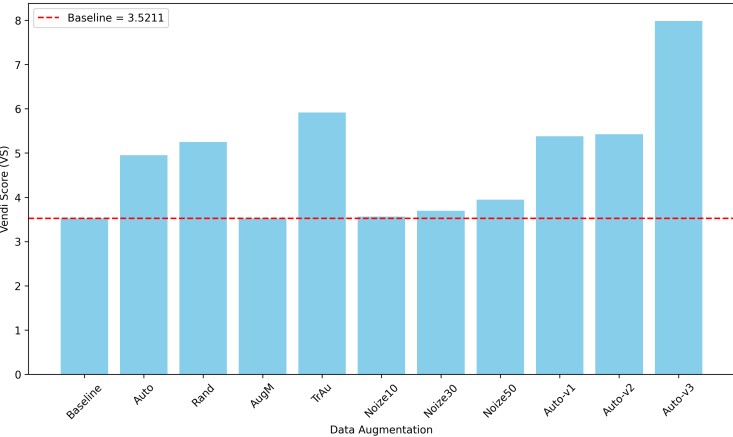

Figure 2: Data diversity in CIFAR-10, as measured by the VS metric, varies across different augmentation methods. The effectiveness of automatic augmentations in enhancing diversity differs, with more advanced augmentations (Auto-v1 to Auto-v3) significantly improving VS outcomes.

**Language Tasks.** We also experiment with BERT ("bert-base-uncased") (Devlin et al., 2019) on the Complaints dataset (Preotiuc-Pietro et al., 2019) (3K tweets, complaints vs. non-complaints). For data augmentation for text data, we use Easy Data Augmentation (EDA) (Wei & Zou, 2019), which includes four rule-based methods: synonym replacement (SR), random insertion (RI), random swap (RS), and random deletion (RD). To generate eight augmented datasets $D_{div}$, we control both dataset-level and sample-level augmentation proportions, setting each to 0.3 and 0.5.

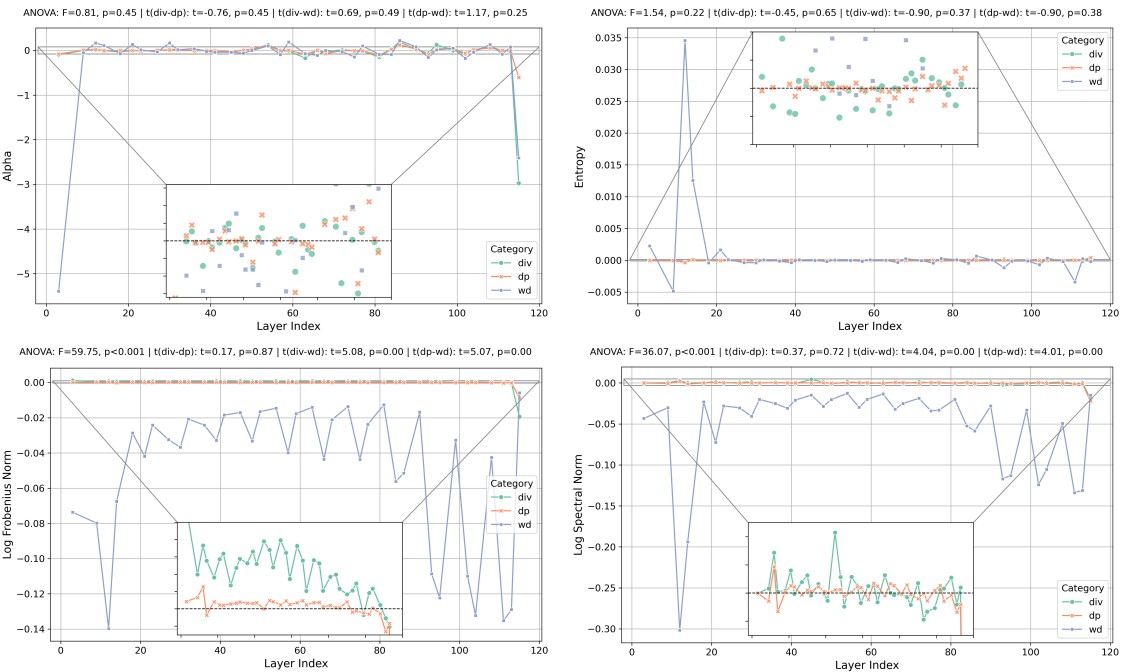

Figure 3: Comparisons of spectral metrics across layers and categories indicate data diversity introduced by augmentations acts as a regularizer, more closely resembling dropout than weight decay. The relative changes in four metrics $\Delta M$ ($\Delta \alpha$, $\Delta$Matrix Entropy, $\Delta$Log Frobenius Norm, $\Delta$Log Spectral Norm) averaged across four datasets in vision tasks for ViT/B-32 fine-tuning. Abbreviations "dp", "wd", and "div" denote dropout, weight decay, and data augmentation, respectively. We provide more visualizations on vision tasks for RN-50 fine-tuning, RN-18 training from scratch, and dataset-wise plots in Figure 9, 10, 13, and 14 in the Appendix E and H.

### 4.2.1 Understanding Model Behavior through Spectral Analysis

We plot a part of the results in Figure 3, which shows the relative changes in four metrics ($\Delta \alpha$, $\Delta$Matrix Entropy, $\Delta$Log Frobenius Norm, $\Delta$Log Spectral Norm), averaged across four datasets and compared to the baseline. Each data point in Figure 3 represents the relative change, calculated as $\Delta M - \Delta M_{baseline}$, with the horizontal line at zero serving as the baseline in each plot. Each figure's title indicates the results of adjusted ANOVA (F-test) and pairwise t-tests. Since the distributions of some categories, particularly those involving dropout and data augmentation, are closely intertwined, we zoom in on the areas around the zero baseline to observe their trends more clearly.

In the Log Frobenius Norm plot, weight decay (wd) reduces Log Frobenius Norm across all layers and exhibits higher overall magnitude reduction compared to the other two categories – dropout and data augmentation. In contrast, both dropout (dp) and data augmentation (div) increase the Frobenius norm in the shallow layers but decrease it in the deeper layers. Notably, in the final layers, both dropout and data augmentation show a sharp decrease, while weight decay displays a rebound. This results in all three techniques exhibiting relative changes in the same direction and of comparable magnitude in the last layers. All of these observations are statistically confirmed by adjusted ANOVA and t-test, and this pattern also appears in the Log Spectral Norm plot.

In contrast, the spectrum patterns in the shape metrics, both $\alpha$ and matrix entropy, do not exhibit clear trends that distinguish between categories. Additionally, statistical tests indicate that these differences are not significant. However, a closer look at the zoomed-in plot reveals that the dropout and data augmentation groups are positioned more closely together.

An interesting and important observation is that the spectral behaviors converge across categories in the final (classification) layer, especially in the Alpha plot, Log Frobenius Norm plot, and Log Spectral Norm plot. For scale-related metrics such as Frobenius Norm and Spectral Norm, weight decay reduces its shrinking effect to a smaller magnitude in the last layer, while dropout and data augmentation cause a more abrupt decrease. Shape metrics like Alpha ($\alpha$) show a sharp decline across all three categories, aligning with the Heavy-Tailed Self-Regularization theory. A smaller $\alpha$ value suggests that the corresponding layer is well-trained. Given the sharp changes observed between the second-to-last and last layer, we also note that the similarity between dropout and data augmentations becomes more evident in this view.

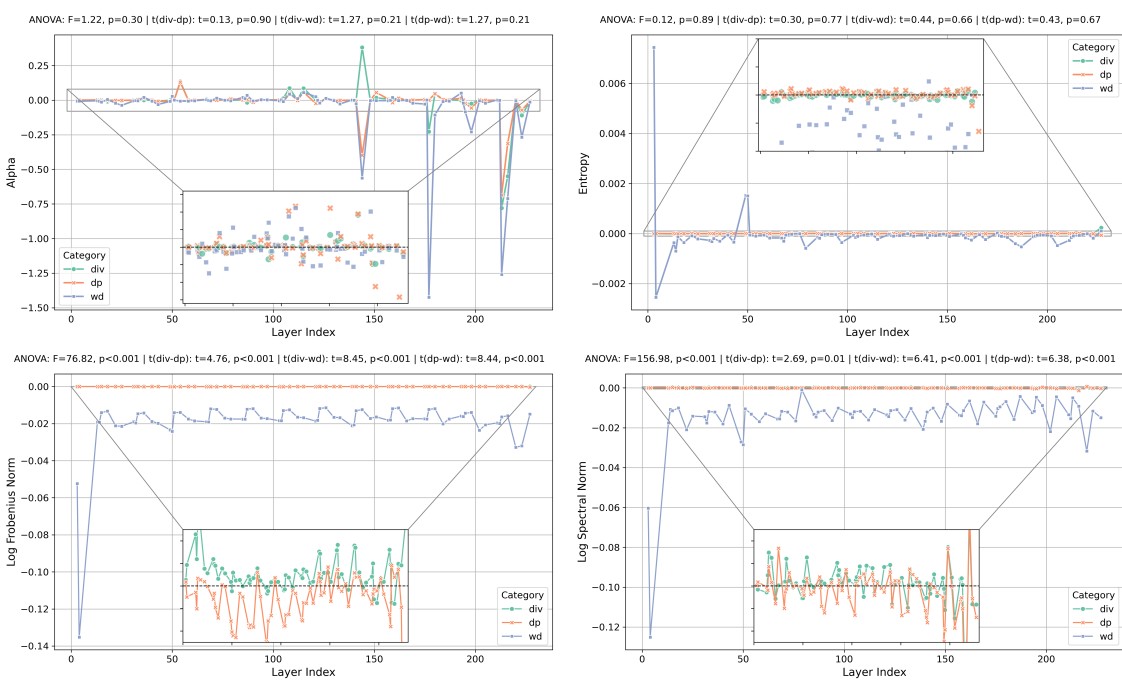

Figure 4: Comparisons of spectral metrics across layers and categories in BERT fine-tuning reveal similar trends in spectral relative changes to those observed in vision tasks. Abbreviations "dp", "wd", and "div" denote dropout, weight decay, and data augmentation, respectively.

Figure 4 presents BERT fine-tuning results on the language task. The four metric plots exhibit trends similar to those in Figure 3 for vision tasks. Notably, the zoom-in entropy plot highlights the closeness between dropout and data augmentations. Additional results for RN-50 fine-tuning and RN-18 training from scratch on vision tasks are provided in Figures 9 and 10 in Appendix E, where consistent patterns with Figure 3 and 4 are also observed.

To disentangle the regularization effects of diverse data from representation effects, we conducted additional experiments on CIFAR-10: (i) norm matching (aligning the Frobenius norm of baseline models with that of models under various regularization schemes to assess predictive performance), (ii) frozen layers (updating only the final classification layers), and (iii) random labels (randomly shuffling the true labels). The results remain consistent with our previous findings, confirming that the regularization induced by data augmentation is feature-independent. Detailed descriptions are provided in the Appendix J.

**Takeaways.** Different regularization methods and data augmentation techniques across vision and language models, whether applied to fine-tuning pretrained models or training from scratch, demonstrate similar spectral characteristics. This suggests a common mechanism by which regularizations reshape the weight parameter space: they reduce the magnitude of the weight parameters $W$ and make the ESD more uniform by increasing its decay rate and boosting its entropy. From comparisons, we also conclude that dropout and data augmentation produce similar spectral patterns of weight matrices.

### 4.2.2 Measuring the Influence of Data Diversity

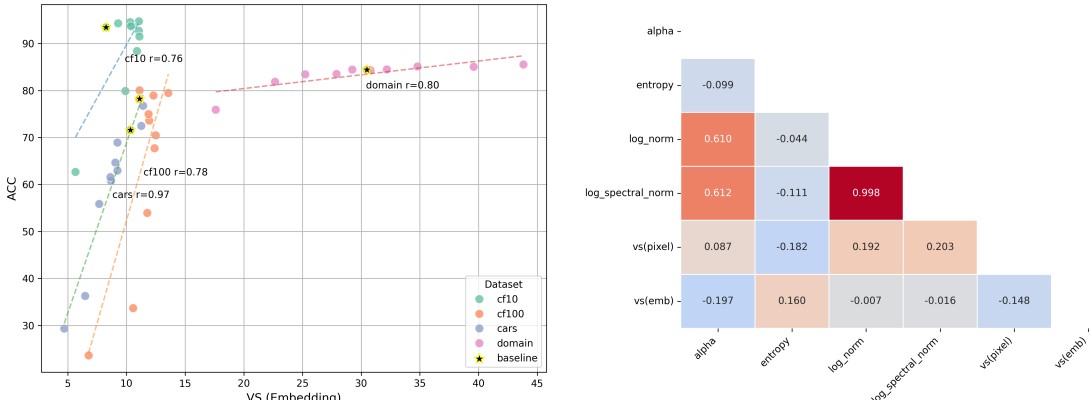

Figure 5: Left: Embedding-based Vendi Score vs Accuracy (Each dot represents one of ten data augmentation techniques as in Fig. 2); Right: Correlation matrix quantifying the relationship between data diversity and spectral metrics.

We examine the extent to which data diversity influences the model's weight matrices and predictive capabilities. As shown in Figure 5 (left), data diversity, measured using the embedding-based Vendi Score, exhibits a positive correlation with model accuracy. The correlation table in Figure 5 (right) indicates that increased data diversity is associated with a reduction in the sum of eigenvalues (log Frobenius norm) and the largest eigenvalue (log spectral norm), while simultaneously leading to an increase in matrix entropy and a decrease in the $\alpha$. This aligns with the theoretical insight from the work (Martin & Mahoney, 2021), which suggests that a well-trained model tends to minimize $\alpha$. We also observe a misalignment between pixel-level and embedding-level[2] measures of data diversity. Therefore, not all data augmentation techniques benefit model training. For pretrained models, only those that effectively increase diversity in the embedding space lead to improved performance, and loss of diversity in the embedding space tends to be associated with degraded model performance.

During training, weight updates are influenced not only by the magnitudes of the weights, but also by their distribution, both of which affect the loss and the curvature of the optimization landscape, as captured by the Hessian. To complement our main analysis, we include a curvature analysis of the loss function landscape in Figure 7 in Appendix C, where we visualize the loss surface by tracking the trace of the Hessian matrix throughout training.

Regularization techniques are well known for their ability to improve generalization. As shown in Tables 2 and 3 in Appendix F, we evaluate the generalization performance of data augmentation in comparison to weight decay and dropout, across both in-distribution (ID) and out-of-distribution (OOD) tasks.

### 4.3 Augmentation by Synthetic Data

Building on the findings on data diversity, we investigate the utility of synthetic data generation as an augmentation tool. Recently, augmenting training data using AI-generated synthetic data has been explored (Zhou et al., 2023; Chen et al., 2024; Ding et al., 2024). We provide insights regarding what synthetic data is beneficial from a diversity perspective. Following the approach of Sahu et al. (2023), we generate synthetic data for the Complaints dataset, described in Section 4.2, near the decision boundary (55% class A, 45% class B) with GPT-4o (Achiam et al., 2023). Six synthetic datasets, denoted as $D_{syn}$, are created by mixing real and synthetic data in ratios of [0.1, 0.3, 0.5, 0.7, 0.9, 1.0], where each ratio indicates the proportion of synthetic data replacing real data to maintain a constant dataset size. To enrich the synthetic data scenarios,

---

[2]We use embedding models that match the fine-tuned model backbones to ensure consistency between data representation and model structure. This enables a more accurate assessment of how input data diversity influences the model's behavior.

we create twelve other datasets with two categories, each with the same ratios as the previous ones. We use the baseline BERT model (fine-tuned with real data) to verify the $D_{syn}$; only samples predicted correctly are used, called $D_{vf}$. Both $D_{syn}$ and $D_{vf}$ have the same size as the original dataset. Then we create $D_{add}$ by adding a certain proportion of synthetic data without removing real data, thereby increasing the overall dataset size.

We observe similar spectral patterns in language models as well, confirming that our findings can generalize beyond vision tasks (see Figures 11 and 12 in Appendix G). To understand the difference between easy data augmentation (EDA) and synthetic data augmentation, we design a weighted Vendi Score, $\widetilde{VS}(X, \tilde{X})$, to capture meaningful diversity, which is defined as

$$\widetilde{VS}(X, \tilde{X}) = \rho(X, \tilde{X}) \cdot VS(\tilde{X}),$$

$$\text{where} \quad \rho(X, \tilde{X}) = \frac{\langle X, \tilde{X} \rangle_F}{\|X\|_F \|\tilde{X}\|_F} \in [-1, 1],$$

Here, $\rho(X, \tilde{X})$ is the cosine similarity between the embedding vector of the original dataset $X$ and the augmented dataset $\tilde{X}$, thus it penalizes augmentations that are misaligned with the original data. This reflects that only meaningful diversity contributes positively to model performance. Figure 6 shows that EDA and synthetic data augmented datasets exhibit different trends in $\widetilde{VS}(X, \tilde{X})$. EDA methods typically achieve higher diversity while maintaining good alignment with the original data, except for Random Deletion, which tends to lose critical information. This is expected, as EDA generates variations grounded in the original dataset. However, excessively increasing data diversity can negatively impact model accuracy.

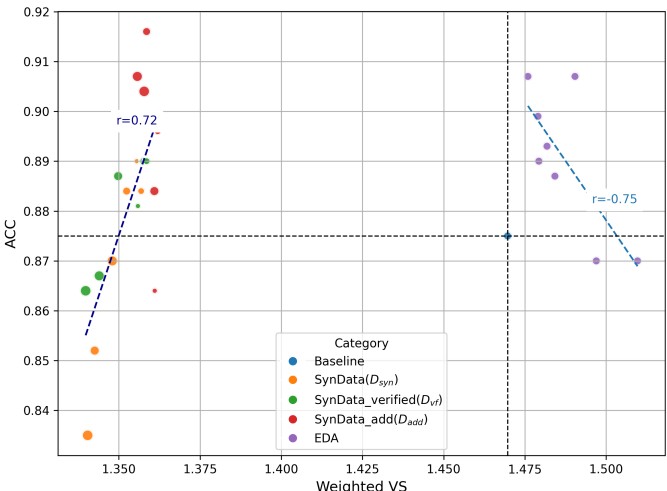

Figure 6: Weighted Vendi Score $\widetilde{VS}(X, \tilde{X})$ vs Accuracy. Dot size reflects the proportion of synthetic data. Purple dots denote EDA shown with equal sizes. The relationship between diversity and model accuracy exhibits distinct trends for traditional versus synthetic data augmentation methods.

In contrast, synthetic datasets often show less alignment, as measured by the scaling factor – cosine similarity $\rho(X, \tilde{X})$, but their performance trends can be better explained by the weighted Vendi Score $\widetilde{VS}(X, \tilde{X})$. For synthetic data, a higher weighted VS value is positively associated with the model performance, showing a correlation of 0.72. When the mixing ratio exceeds 0.5, resulting in a larger loss of diversity, using $D_{syn}$ for model fine-tuning begins to negatively impact model accuracy. However, this mixing ratio threshold can be higher for other synthetic datasets – up to 0.7 for $D_{vf}$ and 1.0 for $D_{add}$. Verified synthetic data tend to exhibit greater similarity to the real data in the embedding space. These observations are consistent with $\widetilde{VS}(X, \tilde{X})$, as lower scores are associated with degraded model performance.

**Takeaways.** The weighted Vendi Score in Figure 6 captures the fundamental difference between traditional data augmentation methods and AI-generated synthetic data augmentation. Synthetic data may introduce

new information not present in the original real dataset, whereas traditional data augmentation methods typically generate variations by recombining or transforming the information that already exists. The weighted Vendi Score serves as a useful metric for assessing what types of synthetic data contribute positively to model training. It also suggests that introducing excessive diversity into the training data, without adding genuinely new information, can be detrimental. Our analysis, grounded in the concept of meaningful diversity, provides a coherent explanation for some previous findings, showing that verified synthetic data (Feng et al., 2024) or synthetic data mixed with real data in an appropriate proportion (Alemohammad et al., 2023; Bertrand et al., 2023) can avoid model collapse.

## 5 Related Work

Data diversity is increasingly recognized for its role in improving generalization, robustness, and fairness, especially in dynamic, real-world settings where training data may not reflect deployment conditions. The rise of synthetic data, domain adaptation, and self-supervised learning has underscored the need to understand how diverse data shapes learned representations. Lopes et al. (2020) examined diversity in data augmentation, defining it via model accuracy and input characteristics measured by conditional entropy. Jung et al. (2025) proposes a gradient-based diversity metric that reliably predicts generalization performance in LLM reasoning tasks, and Zhu et al. (2025) introduces a novel classification-based metric for measuring diversity in LLM-generated synthetic datasets. They showed that data diversity boosts performance, but the mechanisms affecting weight space and learning dynamics remain poorly understood. On the other hand, Zhao et al. (2019) explores a related topic regarding data augmentation and dropout, Shen et al. (2022) reframes data augmentation as feature manipulation, Chidambaram et al. (2023) discusses the learned representations via mixup, and Yang et al. (2023a) shows how consistency-based regularization improves sample efficiency. Our work leverages tools from random matrix theory (RMT) to extend existing understanding by explicitly linking data diversity to weight distribution via spectral characteristics, offering a broader perspective on the implicit regularization effects of augmentation.

The spectral analysis of weight metrics has been applied to various scenarios. Martin & Mahoney (2021) found deep neural networks naturally regularize themselves by shaping their weight matrices, for which they proposed a Heavy-Tailed Self-Regularization (HTSR) theory. Their follow-up work (Martin et al., 2021) used the power-law fitting on weight matrices' spectra to predict a model's quality and generalization ability without needing training or testing data, thus focusing on intrinsic properties. Yang et al. (2023b) applied the HTSR theory in the natural language processing area. All of these prior studies motivated us to investigate the impact of data diversity from the model's weight matrices perspective.

## 6 Conclusion

Data augmentation, by increasing training data diversity, offers a practical approach that not only prevents overfitting but also enhances a model's ability to generalize across varied conditions. Despite documented empirical benefits, the underlying mechanisms through which data augmentation influences learned representations remain unclear. Our study examines how data diversity impacts the weight matrices of DNNs and investigates its relationship to explicit regularization techniques, such as dropout and weight decay.

Leveraging RMT, we analyze the spectral evolution of weight matrices to compare the influence of dropout, weight decay, and data augmentation on their structural properties. Our findings reveal that all three techniques reduce the weight magnitudes and induce similar spectral changes, yet data augmentation exhibits patterns that more closely resemble dropout than weight decay. These results indicate that data augmentation works as an implicit regularizer in DNN training, which has been captured in the spectral patterns of weight matrices. After establishing a clear connection between data diversity and model performance, we further extend our analysis to scenarios involving AI-generated synthetic data augmentation. By applying an adjusted version of the Vendi Score as a measure of dataset diversity, we uncover key differences between the diversity introduced by traditional and synthetic augmentation methods. Our results demonstrate that an appropriate mix of real and synthetic data can significantly enhance model performance.

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

## A    Model Details in Fine-tuned CLIP

The default CLIP-VIT-B32 model does not apply dropouts in the image encoder. To investigate the efforts of dropout on both hidden and output layers, we add dropout layers in both *MultiheadAttention* layers and *feedforward* layers within each transformer block, as well as in the final *classification* layer. In the hidden layers, dropout is applied after the *normalization* layer. For CLIP-ResNet50, we add dropout after each block.

In the tuning process, hyperparameters are set as follows: a learning rate of 1e-5, 5 epochs, and a batch size of 32 are kept the same in all experiments. All model performances are evaluated on real test images. In addition, we train a ResNet-18 from random initialization on the CIFAR-10 dataset in 10 epochs. Our code is implemented based on Pytorch 2.2.1. The pre-trained BERT model is downloaded from the Huggingface library, while pre-trained CLIP models are from OpenAI. Experiments run on Nvidia 3090 GPU.

## B    Eigen-decomposition of Ridge Regression and Dropout: Closed-Form Solution

To make it consistent with the previous expression, we just let $w_{wd}$ as the weight of ridge regression. The objective function of ridge regression can be expressed as

$$L(w_{wd}) = \|y - Xw_{wd}\|^2 + \alpha\|w_{wd}\|^2$$

Taking the derivative of the loss function and equating it to 0,

$$\frac{\partial L}{\partial w_{wd}} = -2X^\top(y - Xw_{wd}) + 2\alpha w_{wd} = 0$$

Simplifying the equation,

$$-X^\top y + (X^\top X + \alpha I)w_{wd} = 0$$

$$(X^\top X + \alpha I)w_{wd} = X^\top y$$

$$w_{wd} = (X^\top X + \alpha I)^{-1}X^\top y$$

Since

$$X^\top X = V\Lambda V^\top$$
$$\alpha I = \alpha VV^\top$$

it is easy to derive

$$X^\top X + \alpha I = V\Lambda V^\top + \alpha VV^\top = V(\Lambda + \alpha I)V^\top$$

Finally, we have

$$w_{wd} = V(\Lambda + \alpha I)^{-1}V^\top X^\top y$$

Similarly, dropout can be interpreted by adding a regularization effect, as in Eq.(2). Therefore, its objective function can be expressed as

$$L(w_{dp}) = \|y - pXw_{dp}\|^2 + p(1-p)\|\Gamma w_{dp}\|^2$$

where $\Gamma = \text{diag}(X^\top X)^{1/2}$.

Taking the derivative of the loss function and equating it to 0,

$$\frac{\partial L}{\partial w_{dp}} = -2pX^\top(y - pXw_{dp}) + 2p(1-p)\Gamma^2 w_{dp} = 0$$

Simplifying the equation,

$$-pX^\top(y - pXw_{dp}) + p(1-p)\Gamma^2 w_{dp} = 0$$

$$pX^\top(y - pXw_{dp}) = p(1-p)\Gamma^2 w_{dp}$$

$$pX^\top y - p^2 X^\top X w_{dp} = p(1-p)\Gamma^2 w_{dp}$$

$$p^2 X^\top X w_{dp} + p(1-p)\Gamma^2 w_{dp} = pX^\top y$$

Factor $w_{dp}$:

$$\left(p^2 X^\top X + p(1-p)\Gamma^2\right) w_{dp} = pX^\top y$$

$$w_{dp} = \left(pX^\top X + p(1-p)\Gamma^2\right)^{-1} pX^\top y$$

$$w_{dp} = \left(p\left(X^\top X + (1-p)\Gamma^2\right)\right)^{-1} pX^\top y$$

$$w_{dp} = \left(\frac{1}{p}\left(X^\top X + (1-p)\Gamma^2\right)^{-1}\right) pX^\top y$$

Then we get the closed-form solution:

$$w_{dp} = \left(X^\top X + (1-p)\Gamma^2\right)^{-1} X^\top y$$

Let $\lambda = (1-p)\Gamma^2$ and do eigen decomposition the same as that in ridge regression, we can obtain

$$w_{dp} = V(\Lambda + \lambda I)^{-1} V^\top X^\top y$$

## C   Curvature Analysis of Loss Function Landscape

In addition to spectral analysis, we include a loss landscape visualization to better understand the implications of these spectral properties manifesting in the learning dynamics. The landscape reflects how the model moves through parameter space, with its curvature captured by the Hessian. Examining the Hessian reveals how the magnitude and distribution of $W$ affect loss changes and shape the trace. Figure 7 illustrates three distinct training stages.

**Stage 1: Rapid updates.** After the initial phase (iterations 0–100), where the model passes through a saddle point, training enters a rapid update stage (iterations 100–245). During this period, the weight matrix $W$ resides in a sharp region of the loss landscape, marked by high curvature and large second derivatives, leading to increased Hessian trace. Regularizers like dropout (rate 0.1) and weight decay (5e-3) show elevated trace values, while data augmentations such as TrivialAugment (TrAu) and Gaussian noise (Noise_05) yield even higher traces, indicating more substantial updates and stronger learning signals.

**Stage 2: Transition between sharp and flat regions.** As training continues, changes in $W$ diminish, reflecting a transition to flatter regions. The Hessian trace for dropout and weight decay drops below the baseline, indicating a smoothing effect. Correspondingly, data augmentations still yield higher traces, though the gap narrows, suggesting continued exploration with increasing stability.

**Stage 3: Convergence to a flat region.** In the final phase, the model settles into a low-curvature region. Dropout and weight decay trace values remain below the baseline, guiding the model toward flatter minima. Trace values for augmentation methods also decline, aligning more closely with the baseline, reflecting their stabilizing, regularization-like role in promoting generalization.

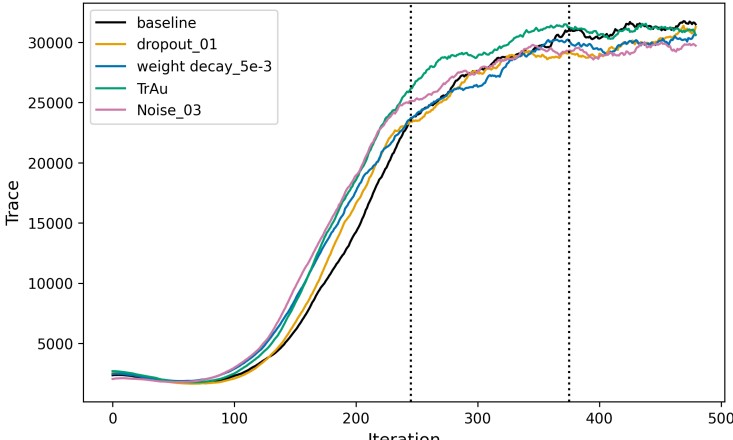

Figure 7: Visualization of loss function curvature characteristic derived from fine-tuning a pre-trained ResNet-18 on Home-office data. We train the baseline, all levels of regularization, and data augmentations in 480 iterations, 5 epochs. For each iteration, we compute the loss curvature and report the smoothed trace of the corresponding Hessian matrix (Yang et al., 2024) using Hutchinson's method (Hutchinson, 1989) implemented in PyHessian (Yao et al., 2020). Three distinct stages of the training process are presented in these plots.

# D Terminology and Interpretation of Metric Changes

## D.1 Detailed Descriptions of Terminology

For a clearer understanding, we revisit several concepts introduced in Section 3.

**Empirical Spectral Distribution (ESD).** The Empirical Spectral Distribution describes the distribution of eigenvalues of a matrix. Given a matrix $X \in \mathbb{R}^{n \times n}$ with eigenvalues $\lambda_1, \lambda_2, \ldots, \lambda_n$, the ESD is defined as:

$$F_n(\lambda) = \frac{1}{n} \sum_{i=1}^{n} \mathbf{1}_{\{\lambda_i \leq \lambda\}},$$

where $\lambda \in \mathbb{R}$ is a real-valued threshold, and $\mathbf{1}$ is the indicator function. ESD is widely used in random matrix theory, signal processing, and machine learning to analyze the structure and stability of data matrices. Large eigenvalues typically indicate meaningful signals, while smaller eigenvalues often correspond to noise.

**Power Law (PL).** A power law is a type of statistical distribution where the probability of an event decreases polynomially with its size:

$$p(x) \sim x^{-\alpha},$$

for some exponent $\alpha > 0$. In machine learning, power-law behavior in weight matrices is often associated with strong generalization properties. In this paper, we apply PL to the ESD of weight matrices to characterize the tail behavior of the ESD (Eq. 3). According to the theory proposed by Martin & Mahoney (2021), the tail properties of weight matrices reflect the knowledge learned by the model.

**Noise/Signal Decomposition of ESD.** Based on insights from the PL of the ESD, the weight matrix $W$ can be expressed as $W \simeq W_{rand} + W_{signal}$, where $W_{rand}$ represents "noise" and $W_{signal}$ represents "signal". We transform the ESD in Figure 1 to a logarithmic scale to better visualize both the bulk and tail properties, and to illustrate how $\alpha$ reflects the tail behavior of the ESD. As shown in each plot in Figure 8, $\lambda_{min}$ separates the ESD into bulk (left) and tail (right) regions, corresponding to $W_{rand}$ and $W_{signal}$, respectively. A power law is then fitted to the tail to estimate the value of $\alpha$.

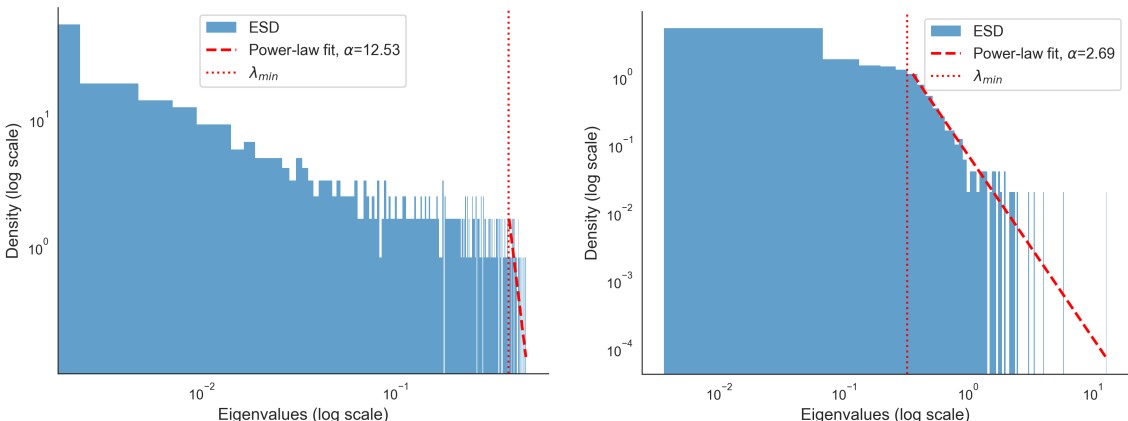

Figure 8: Plots of log ESDs from feedforward layers of a fine-tuned CLIP model on CIFAR-100. The bulk and tail parts are separated by the selected $\lambda_{min}$. Clearly, $\alpha$ guides the shape of the distribution. A large $\alpha$ (**Left** panel) causes the eigenvalues to be more evenly spread, while a small $\alpha$ (**Right** panel) leads to a more "heavy-tailed" distribution of the eigenvalue spectrum.

**Isotropy.** Isotropy refers to properties that are identical in all directions. A random vector $x \in \mathbb{R}^n$ is isotropic if its covariance matrix is proportional to the identity:

$$\mathbb{E}[xx^T] = \sigma^2 I_n.$$

Isotropic noise or embeddings have no preferred direction, simplifying analysis and improving robustness.

### D.2 Interpretations of changes in scale and shape metrics

Table 1: Explaining how changes in scale and shape metrics influence the behavior of the weight matrix and its spectral density.

| Metric | Change | $W$ | ESD |
|---|---|---|---|
| $\alpha$ | ↑ | Less complex $W$, fewer directions dominate. | A sharper drop-off in ESD, less heavy-tailed |
| | ↓ | More complexity, more dominant directions. | A slower decay in ESD, more eigenvalues with large magnitudes |
| Matrix Entropy | ↑ | More diverse and less structured weight patterns | More uniform distribution of $\lambda$ across the spectrum |
| | ↓ | More organized and meaningful weight structures | More concentrated around a few large $\lambda$ |
| Frobenius Norm | ↑ | Larger weight values overall | Larger $\lambda$ in total |
| | ↓ | Smaller weight values | Overall $\lambda$ shrink |
| Spectral Norm | ↑ | At least one dominant direction in $W$ | More outliers in the long tail |
| | ↓ | The maximum influence of any single direction is reduced | Smaller maximum $\lambda$, more compact distribution |

Table 1 describes how changes in scale and shape metrics relate to the structure of the weight matrix $W$ and its spectral properties (ESD). In general, decreases in shape metrics indicate more structured and more compact spectra, while increases in scale metrics reflect greater magnitude in $W$. This table highlights how spectral and norm-based metrics can capture key aspects of model complexity and weight distribution.

## E   Spectral Analysis for Different Models and Tasks

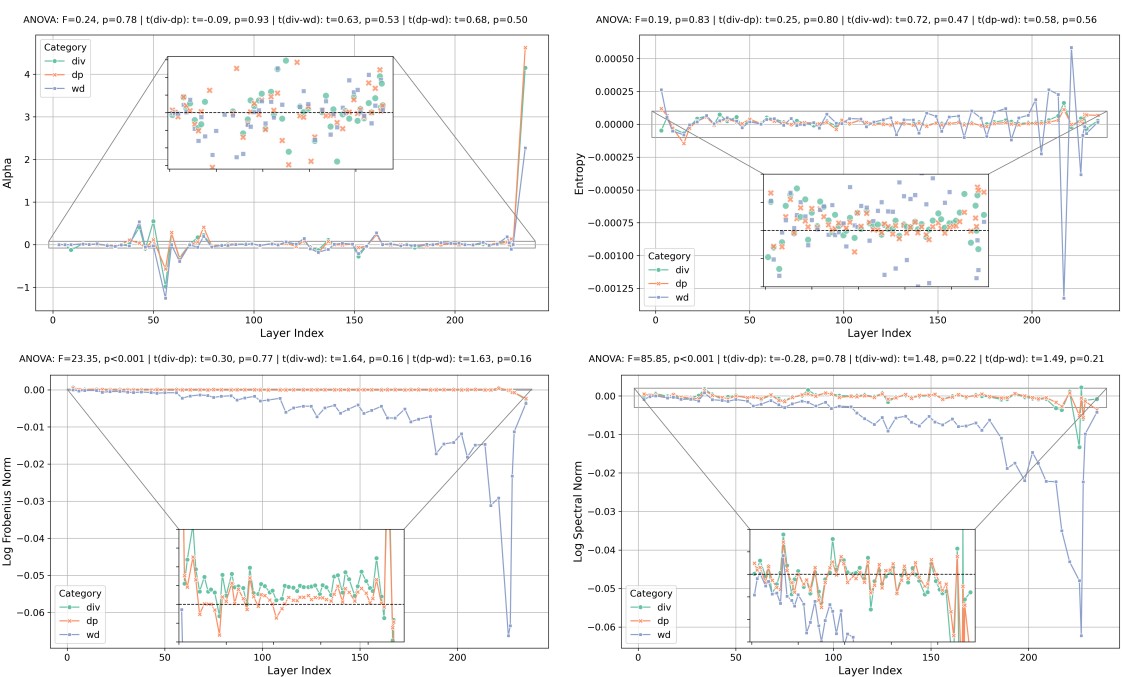

Figure 9: Comparison of spectral metrics across layers and categories. The relative changes in four metrics $\Delta M$ averaged across four datasets in vision tasks for RN-50 fine-tuning.

We present visualizations of the relative changes in four metrics for RN-50 fine-tuning in Figure 9 and RN-18 training from scratch in Figure 10 on vision tasks. Similar patterns to those shown in Figure 3 are also observed in these visualizations. This suggests that our conversations are not limited to specific models or tasks. It further validates our conclusion that the mechanism of data diversity serves as an implicit regularization.

## F   ID & OOD Performance

We evaluate the performance of different regularization approaches and data augmentation techniques on both in-distribution (ID) and out-of-distribution (OOD) tasks by fine-tuning CLIP on the DomainNet (Peng et al., 2019) and Office-Home (Venkateswara et al., 2017) datasets. The baseline models are trained on "real" categories (randomly select 10000 instances for DomainNet, covering 34 classes), while data augmentation techniques are applied to the "real" training data for model training as a comparison. We assess the performance of ID tasks using the "real" test data. For OOD tasks, we evaluate trained models on stylized images, including "clipart", "infograph", "inpainting" and "sketch" styles from DomainNet and "clipart", "painting" and "art" styles from Office-Home dataset. All model parameters remain consistent with those used in previous sections. Expected Calibration Error (ECE), which quantifies how well a model's predicted probabilities (confidence) align with its actual outcomes (accuracy) (Guo et al., 2017), has been studied as a benefit of data augmentations (Zhang, 2017; Thulasidasan et al., 2019). The smaller the ECE is, the better the model is calibrated. We evaluate and confirm their performance as well.

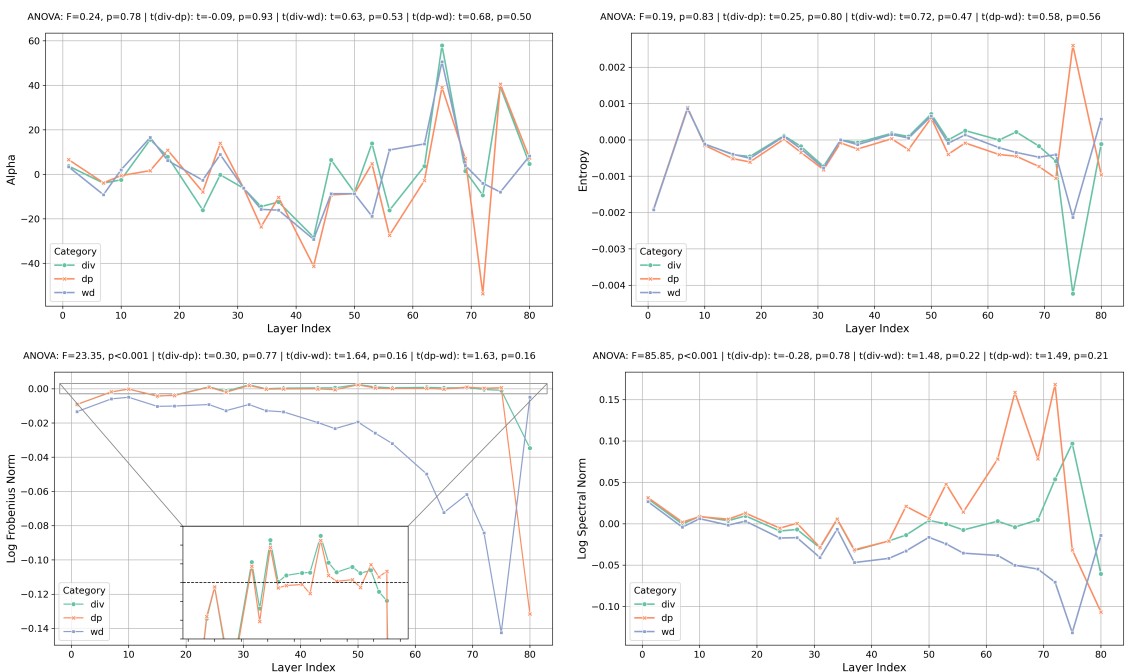

Figure 10: Comparison of spectral metrics across layers and categories. The relative changes in four metrics $\Delta M$ for RN-18 training on scratch on CIFAR-10.

Table 2: The *Top-1* accuracy evaluations for both in-distribution (ID) and out-of-distribution (OOD) tasks.

| | **DomainNet** | | | | | | **Office-Home** | | | | |
|---|---|---|---|---|---|---|---|---|---|---|---|
| | ID | OOD | | | | AVG | ID | OOD | | | AVG |
| | Acc | Clipart | Infograph | Painting | Sketch | Acc | Acc | Clipart | Product | Art | Acc |
| | | Acc | Acc | Acc | Acc | | | Acc | Acc | Acc | |
| Baseline | 93.75 | 56.12 | 31.47 | 60.11 | 56.08 | 50.95 | 74.31 | 37.34 | 65.24 | 52.78 | 51.79 |
| DP01 | 93.69 | 55.86 | 30.96 | 59.06 | 57.85 | 50.93 | 72.36 | 39.02 | 66.21 | 55.05 | 53.43 |
| DP03 | 93.25 | 55.78 | 30.28 | 55.08 | 56.59 | 49.43 | 74.54 | 37.30 | 66.66 | 56.65 | 53.54 |
| DP05 | 93.60 | 56.42 | 30.82 | 57.55 | 56.34 | 50.28 | 74.31 | 38.79 | 65.71 | **57.23** | 53.91 |
| DP07 | 93.32 | 56.24 | 30.79 | 57.78 | 56.83 | 50.41 | 70.76 | 39.27 | 62.15 | 55.42 | 52.28 |
| WD1e-05 | 93.48 | 56.38 | 31.56 | 58.96 | 57.45 | 51.09 | 75.0 | 37.69 | 67.61 | 54.35 | 53.22 |
| WD5e-05 | **93.85** | 57.71 | 31.70 | 60.48 | 58.73 | 52.15 | 72.48 | 37.66 | 66.37 | 54.27 | 52.77 |
| WD1e-4 | 93.23 | **61.41** | 33.03 | 60.36 | **64.49** | **54.82** | 74.77 | 37.30 | 65.49 | 55.87 | 52.89 |
| WD5e-4 | 93.55 | 56.73 | 30.99 | 60.65 | 62.59 | 52.74 | 73.05 | 36.50 | 66.64 | 56.98 | 53.37 |
| WD1e-3 | 93.35 | 60.77 | 33.43 | 61.14 | 60.93 | 54.07 | 73.85 | 38.92 | 67.40 | 56.41 | 54.24 |
| WD5e-3 | 93.39 | 55.52 | 29.69 | 57.55 | 58.60 | 50.34 | 73.85 | 37.32 | 65.78 | 56.41 | 53.17 |
| Auto | 92.91 | 56.05 | 30.00 | 58.51 | 57.74 | 50.57 | 72.94 | 37.00 | 66.01 | 54.31 | 52.44 |
| Rand | 93.35 | 55.97 | 31.13 | 59.43 | 62.54 | 52.27 | 73.39 | 35.33 | 66.39 | 54.92 | 52.21 |
| AugM | **93.73** | 57.79 | 29.83 | 57.85 | 57.36 | 50.71 | 74.20 | 36.63 | 66.30 | 54.64 | 52.52 |
| TrAu | 93.35 | 60.13 | 32.27 | 61.02 | **62.85** | **54.07** | 72.82 | 37.25 | 67.00 | 55.05 | 53.10 |
| Noise10 | 93.23 | 59.03 | 32.10 | **62.05** | 58.84 | 53.01 | 74.08 | **40.53** | 67.00 | 56.16 | **54.56** |
| Noise30 | 93.05 | 57.07 | **33.09** | 59.30 | 60.81 | 52.57 | 72.71 | 38.67 | **68.12** | 55.13 | 53.97 |
| Noise50 | 93.69 | **60.58** | 32.69 | 59.80 | 60.79 | 53.47 | **75.57** | 40.16 | 66.57 | **56.41** | 54.38 |
| Auto-v1 | 93.03 | 58.99 | 32.01 | 55.98 | 57.36 | 51.09 | 74.89 | 38.51 | 66.46 | 55.54 | 53.50 |
| Auto-v2 | 93.53 | 59.26 | 31.59 | 60.29 | 60.68 | 52.96 | 72.36 | 38.01 | 66.66 | 54.59 | 53.09 |
| Auto-v3 | 93.62 | 58.81 | 31.33 | 60.11 | 61.41 | 52.91 | 74.54 | 37.82 | 66.59 | 54.64 | 53.02 |

**Result.** Table 2 presents the results from fine-tuning CLIP with a ResNet-50 backbone. The left table shows the results for training and testing on the DomainNet dataset, while the right table displays the results for the Office-Home dataset. "AVG" denotes the average accuracy across all OOD tasks. This table shows

Table 3: Expected Calibration Error (ECE) evaluations for both in-distribution (ID) and out-of-distribution (OOD) tasks.

| | DomainNet | | | | | | Office-Home | | | | |
| | ID | OOD | | | | AVG | ID | OOD | | | AVG |
| | ECE | Clipart ECE | Infograph ECE | Painting ECE | Sketch ECE | ECE | ECE | Clipart ECE | Product ECE | Art ECE | ECE |
|---|---|---|---|---|---|---|---|---|---|---|---|
| Baseline | 0.016 | 0.179 | 0.320 | 0.136 | 0.168 | 0.201 | **0.030** | 0.162 | 0.028 | 0.068 | 0.086 |
| DP01 | 0.018 | 0.194 | 0.334 | 0.148 | 0.184 | 0.215 | 0.043 | 0.181 | 0.029 | 0.050 | 0.087 |
| DP03 | 0.019 | 0.197 | 0.318 | 0.155 | 0.187 | 0.214 | 0.054 | 0.150 | 0.030 | 0.035 | 0.072 |
| DP05 | 0.020 | 0.183 | 0.332 | 0.167 | 0.184 | 0.217 | 0.068 | 0.091 | 0.037 | **0.028** | **0.052** |
| DP07 | 0.019 | 0.207 | 0.310 | 0.156 | 0.191 | 0.216 | 0.082 | **0.066** | 0.055 | 0.060 | 0.060 |
| WD1e-05 | 0.021 | 0.205 | 0.376 | 0.154 | 0.198 | 0.233 | 0.034 | 0.121 | **0.021** | 0.058 | 0.067 |
| WD5e-05 | 0.018 | 0.168 | 0.329 | 0.132 | 0.137 | 0.192 | 0.037 | 0.136 | 0.032 | 0.059 | 0.076 |
| WD1e-4 | 0.026 | 0.151 | 0.354 | 0.154 | 0.128 | 0.197 | 0.050 | 0.166 | 0.026 | 0.061 | 0.084 |
| WD5e-4 | 0.024 | 0.193 | 0.340 | 0.141 | 0.154 | 0.207 | 0.043 | 0.171 | 0.029 | 0.046 | 0.082 |
| WD1e-3 | 0.023 | 0.143 | 0.303 | 0.141 | 0.142 | 0.182 | 0.038 | 0.158 | **0.021** | 0.051 | 0.077 |
| WD5e-3 | **0.015** | 0.196 | 0.400 | 0.165 | 0.153 | 0.229 | 0.041 | 0.177 | 0.036 | 0.058 | 0.09 |
| Auto | 0.025 | 0.197 | 0.416 | 0.164 | 0.194 | 0.243 | 0.040 | 0.192 | 0.035 | 0.066 | 0.098 |
| Rand | 0.022 | 0.195 | 0.322 | 0.125 | 0.127 | 0.192 | 0.056 | 0.205 | 0.026 | 0.069 | 0.100 |
| AugM | 0.021 | 0.198 | 0.373 | 0.157 | 0.187 | 0.229 | 0.044 | 0.148 | 0.029 | 0.050 | 0.076 |
| TrAu | 0.023 | **0.135** | 0.323 | **0.116** | **0.119** | **0.173** | 0.034 | 0.147 | 0.041 | 0.064 | 0.084 |
| Noise10 | 0.021 | 0.169 | 0.354 | 0.130 | 0.158 | 0.203 | 0.046 | 0.141 | **0.025** | 0.062 | **0.076** |
| Noise30 | 0.024 | 0.207 | 0.309 | 0.141 | 0.156 | 0.203 | 0.041 | 0.156 | 0.036 | 0.057 | 0.081 |
| Noise50 | **0.016** | 0.142 | **0.267** | 0.149 | 0.145 | 0.176 | 0.049 | 0.139 | 0.027 | 0.068 | 0.078 |
| Auto-v1 | 0.020 | 0.178 | 0.338 | 0.173 | 0.188 | 0.219 | 0.056 | 0.168 | 0.026 | 0.079 | 0.091 |
| Auto-v2 | 0.022 | 0.136 | 0.361 | 0.122 | 0.128 | 0.187 | 0.046 | 0.168 | 0.033 | 0.070 | 0.09 |
| Auto-v3 | 0.021 | 0.169 | 0.362 | 0.140 | 0.153 | 0.206 | **0.034** | 0.137 | 0.035 | 0.065 | 0.077 |

that data augmentation can yield comparable performance for in-distribution evaluation, and it significantly improves the performance on out-of-distribution datasets, compared with the baseline, dropout, and weight decay. On the other hand, data augmentation demonstrates superior performance in model calibration on DomainNet for both ID and OOD scenarios. However, on the Office-Home dataset, dropout achieves the best performance, despite data augmentation outperforming the baseline as shown in Table 3.

## G   More on Spectral analysis for BERT

The upper panel of Figure 11 illustrates the evolution of spectral measurements across layers among regularization and data augmentations. The consistent spectral patterns observed support the generalizability of our findings from Section 4. The lower panel of Figure 11 presents a comparison among the three categories of synthetic datasets, revealing that their impacts on the model's weight space are largely comparable.

Figure 12 shows the average changes in four spectral metrics across all layers under different scenarios. The results highlight a notable similarity between the effects of dropout and data augmentations, including both traditional methods like EDA and synthetic augmentation techniques, on model representations.

Table 4 summarizes the average values across categories relative to the baseline, providing insights into the diversity introduced by traditional data augmentations, such as EDA, and synthetic data augmentations, as well as their corresponding impact on model accuracy. The results also reveal similarities within augmentation groups, helping to explain why the weighted Vendi Score effectively captures the differences in diversity between them and correlates with performance outcomes.

## H   More on Visualization for Spectral Analysis

Dataset-level spectral visualizations across network layers are shown in Figure 13 and Figure 14. The observed patterns are consistent and robust across different datasets and model architectures. We found that

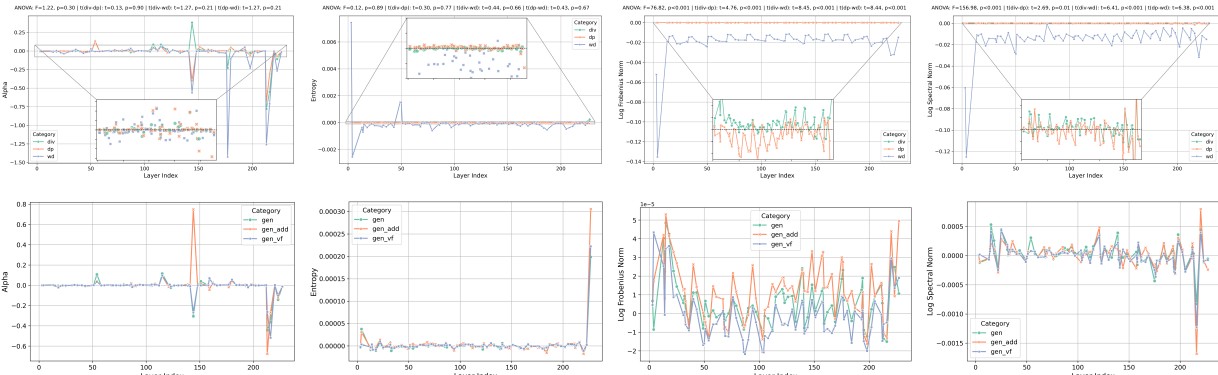

Figure 11: Spectral Metrics Changes through layers for BERT. **1st line:** We present the results for Dropout, Weigh Decay, and EDA; **2nd line:** Comparisons among synthetic datasets

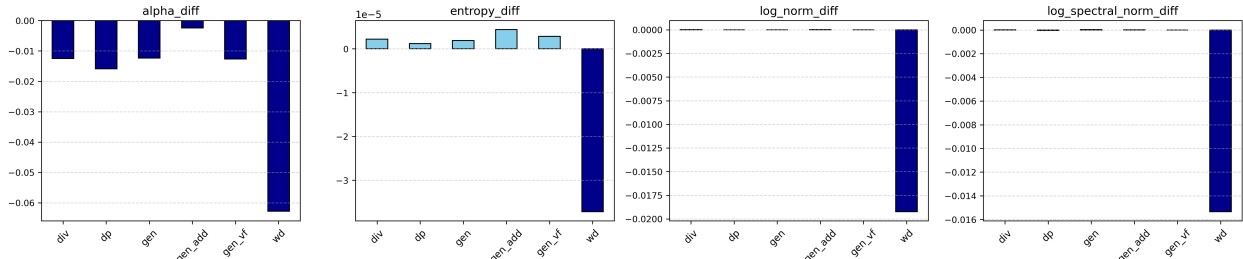

Figure 12: Average Changes in Spectral Metrics for BERT on TC dataset

Table 4: Comparison of Metrics Across Categories

| Category | VS | Accuracy | $\rho$ | $\widetilde{\text{VS}}$ |
|---|---|---|---|---|
| Baseline | 1.4697 | 0.8750 | 1.0000 | 1.4697 |
| $D_{div}$ | 1.5037 | 0.8904 | 0.9891 | 1.4872 |
| $D_{gen}$ | 1.4636 | 0.8692 | 0.9219 | 1.3493 |
| $D_{add}$ | 1.4740 | 0.8952 | 0.9222 | 1.3593 |
| $D_{vf}$ | 1.4659 | 0.8798 | 0.9216 | 1.3510 |

the patterns identified in the experimental section hold consistently across different datasets: regularization tends to reduce the magnitude of the Log Frobenius Norm and Log Spectral Norm and increase matrix entropy. The change in $\alpha$ varies across different backbone architectures between CLIP-ViT-B/32 and CLIP-ResNet-50, but remains consistent across datasets within the same model structure.

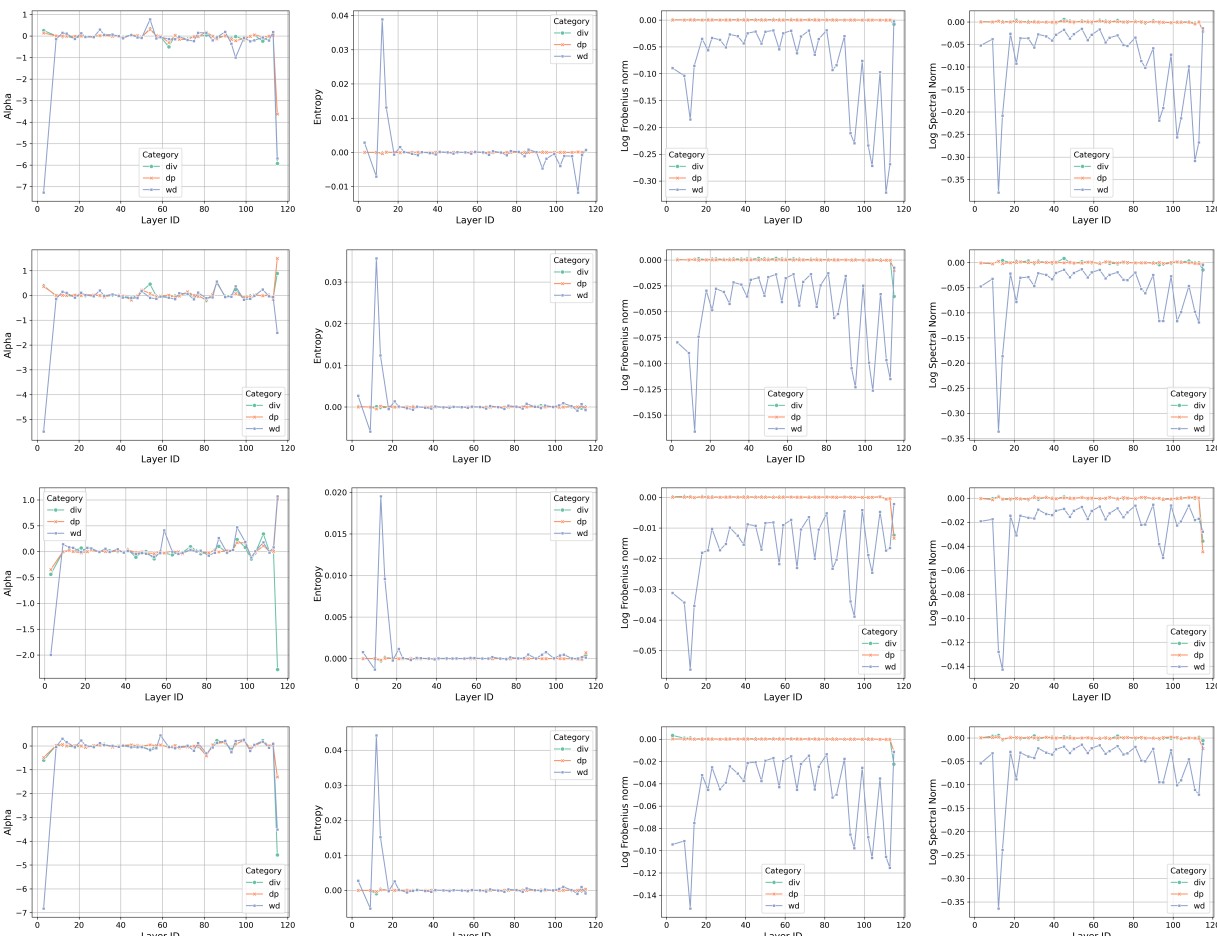

Figure 13: Comparison of spectral metrics across layers and categories for CLIP-ViT-B32. The relative changes in four metrics $\Delta M$ averaged across four datasets. **1st line:** CIFAR-10; **2nd line:** CIFAR-100; **3rd line:** Stanford Cars; **4th line:** DomainNet.

## I Spectral Analysis with Combinations

Since training or fine-tuning on diverse datasets often involves regularization techniques such as dropout and weight decay, we conducted a series of experiments combining different levels of dropout (0.1, 0.5) and weight decay (1e-5, 5e-3) with 10 data augmentations on the Stanford Cars dataset. For these experiments, we fine-tuned the CLIP-ViT-B32 model to evaluate the joint effects of data augmentation and regularization. Figure 15 presents the relative changes in four spectral metrics compared to using data augmentation alone. The top panel shows the average changes across all layers, while the bottom panel focuses specifically on changes in the last layer.

Our results confirm that the combined application of dropout or weight decay enhances the regularization effects of data augmentations (The upper Figure 15), evidenced by increased matrix entropy and decreased $\alpha$ in shape metrics, along with reductions in scale metrics such as the log Frobenius norm and log spectral norm. These findings are consistent with the main conclusions of our paper.

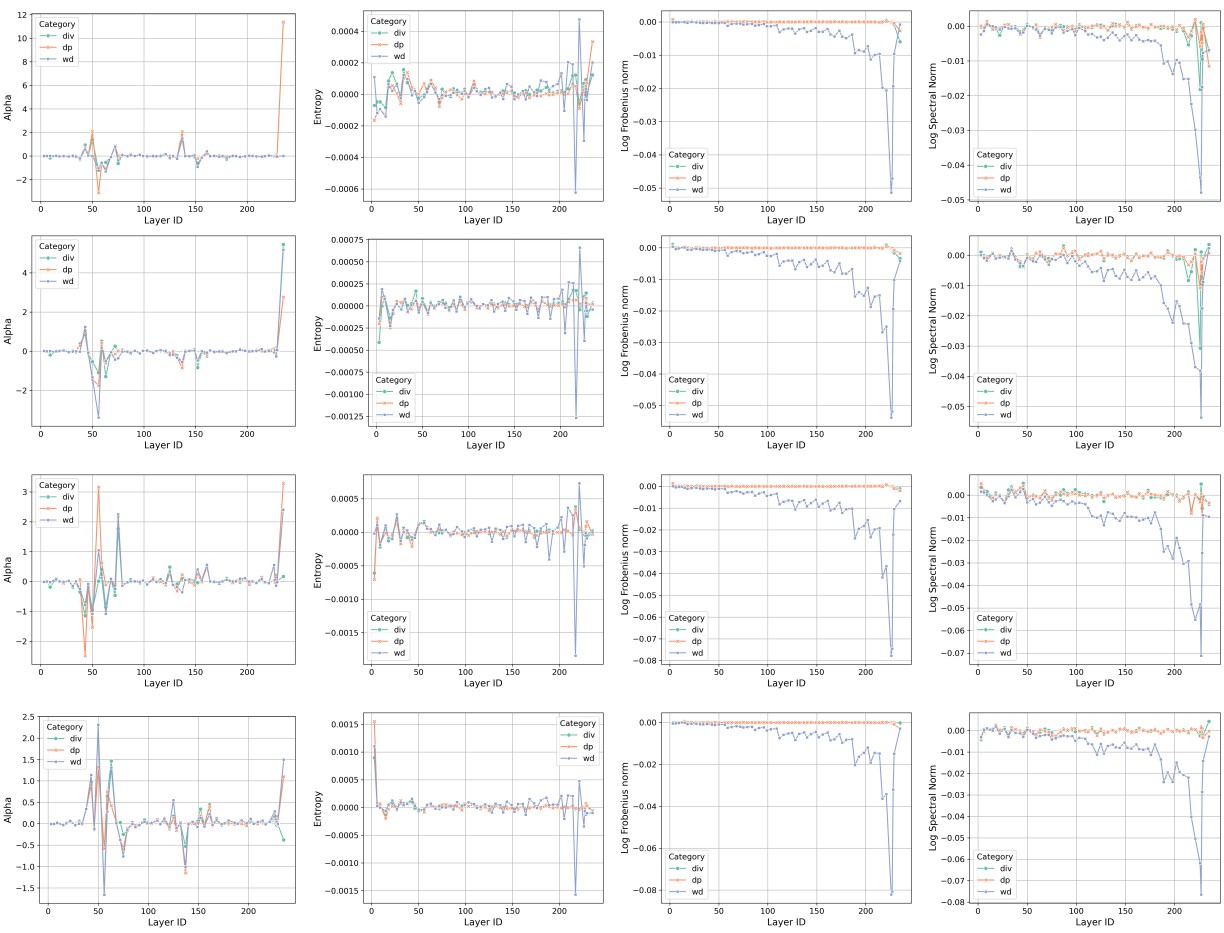

Figure 14: Comparison of spectral metrics across layers and categories for CLIP-RN-50. The relative changes in four metrics $\Delta M$ averaged across four datasets; **1st line:** CIFAR-10; **2nd line:** CIFAR-100; **3rd line:** Stanford Cars; **4th line:** DomainNet.

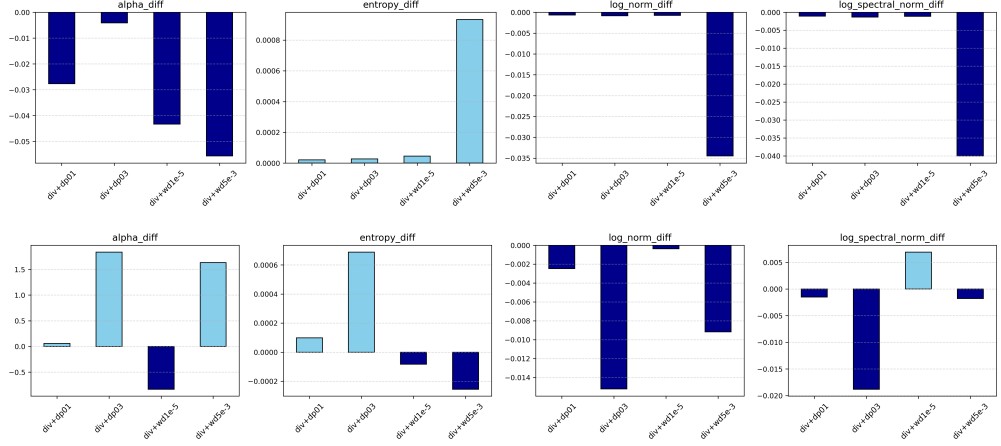

Figure 15: Impact of Combined Regularization on Spectral Metrics Relative to Data Augmentation

# J  Additional Experiments

## J.1  Norm Matching

Let the baseline model weights be $W_{\text{base}}$ with Frobenius norm $\|W_{\text{base}}\|$, and the target model weights be $W_{\text{target}}$ with norm $\|W_{\text{target}}\|$. The scaling factor is defined as

$$s = \frac{\|W_{\text{target}}\|}{\|W_{\text{base}}\|},$$

and the scaled weights are given by

$$W_{\text{base\_scaled}} = s \cdot W_{\text{base}}.$$

This ensures that the baseline model is matched in terms of norm with three different approaches. Then we use matched or weight-rescaled baseline models to predict on the test set.

Table 5: Norm matching on CIFAR-10 for cargoriy-wise $\Delta$ log Frobenius norm vs $\Delta$ Accuracy. A larger reduction in the Frobenius norm during training is associated with greater accuracy changes.

| Category | $\Delta$ log Frobenius norm | $\Delta$ Accuracy |
|---|---|---|
| data augmentations | -0.000028 | -0.005000 |
| dropout | -0.000070 | 0.002500 |
| weight decay | -0.082867 | -13.538333 |

The results in Table 5 show that weight decay–based norm matching causes the most substantial accuracy changes, while dropout and data augmentation lead to only minor deviations from the baseline. These findings are consistent with the greater magnitude of weight norm changes induced by weight decay compared to the smaller effects of dropout and data augmentation. (refer to Figures 3 and 4). Overall, this experiment suggests that larger differences in Frobenius norm significantly affect model prediction performance, confirming the importance of weight norm in model generalization and highlighting the regularization effect provided by data diversity.

## J.2  Frozen Image Encoders

In this experiment, we freeze the image encoder in CLIP, and only the weights in the last classification layer are allowed to be updated.

Table 6: Category-wise metric changes in classification layers on CIFAR-10

| Category | $\Delta\alpha$ | $\Delta$ Entropy | $\Delta$ log Frobenius norm | $\Delta$ log spectral norm |
|---|---|---|---|---|
| data augmentations | -9.757364 | -0.003237 | -0.059237 | -0.001415 |
| dropout | -10.633206 | 0.004380 | -0.083929 | -0.068499 |
| weight decay | -13.887448 | -0.000680 | -0.011179 | 0.018215 |

As Table 6 shows, the same directional changes in both alpha and the Frobenius norm are observed across all three approaches. Moreover, the directional shifts in alpha, Frobenius norm, and spectral norm were closely aligned for dropout and data augmentations, suggesting that these methods influence the weights in a similar manner.

## J.3 Random Labels

We randomly shuffle 30% and 50% of labels in the training set of CIFAR-10.

Table 7: 30% random labels results for category-wise differences with ANOVA $p$-value.

| Category | $\Delta\alpha$ | $\Delta$ Entropy | $\Delta$ log Frobenius norm | $\Delta$ log spectral norm |
|---|---|---|---|---|
| data augmentations (div) | -0.018033 | -0.000009 | -0.000274 | -0.000151 |
| dropout (dp) | 0.006047 | 0.000006 | -0.000175 | -0.000372 |
| weight decay (wd) | -0.226203 | 0.000475 | -0.087043 | -0.092725 |
| ANOVA ($p$) | 0.2671 | 0.8581 | **0.0000** | **0.0000** |

Pairwise t significant differences ($p < 0.05$): $\Delta$ log Frobenius norm: div vs wd, dp vs wd.
$\Delta$ log spectral norm: div vs wd, dp vs wd.

Table 8: 50% random labels results for category-wise differences with ANOVA $p$-value.

| Category | $\Delta\alpha$ | $\Delta$ Entropy | $\Delta$ log Frobenius norm | $\Delta$ log spectral norm |
|---|---|---|---|---|
| data augmentations (div) | -0.050712 | -0.000022 | -0.000460 | -0.000105 |
| dropout (dp) | -0.073727 | -0.000008 | -0.000260 | -0.000147 |
| weight decay (wd) | -0.277604 | 0.000125 | -0.096175 | -0.100430 |
| ANOVA ($p$) | 0.3696 | 0.9873 | **0.0000** | **0.0000** |

Pairwise t significant differences ($p < 0.05$): $\Delta$ log Frobenius norm: div vs wd, dp vs wd.
$\Delta$ log spectral norm: div vs wd, dp vs wd.

The results consistently show that all three methods (weight decay, dropout, and data augmentation) reduce both the Frobenius norm and spectral norm, with the relative magnitude of Frobenius norm shrinkage on average following the order: weight decay > data augmentation > dropout.

At the 0.3 corruption level (Table 7), scale-related metrics such as alpha and entropy exhibit slightly different directional changes across methods, but the adjusted ANOVA test confirms that these differences in alpha and entropy are not statistically significant, aligning with prior observations in the paper. At the 0.5 corruption level (Table 8), a similar pattern emerges: all three methods reduce Frobenius norm, spectral norm, and alpha, while entropy again diverges in directionality: this time with dropout aligning more closely with data augmentation, but differing from weight decay. These differences in entropy are also statistically non-significant according to the adjusted ANOVA test, consistent with reported findings in the paper.

Overall, the frozen image encoders and random labels experiments suggest that the regularization effect of data diversity introduced through augmentation is independent of feature learning and behaves in a manner similar to dropout, particularly in terms of the magnitude of norm decrease, whereas weight decay exerts a stronger influence. This confirms previous findings, and its regularization is different from representation effects.

# K  Adjusted Statistical Testing

The adjusted statistical tests mentioned in Section 4 are applied by following: for each time series data with sample size $n$, we calculate lag-1 autocorrelation $\rho$. Then we recompute the effective sample size $n_{\text{adj}}$.

$$n_{\text{adj}} = n * \frac{1 - \rho}{1 + \rho}$$

### K.1 Adjusted ANOVA Test

Null hypothesis: $\Delta M$ for all three different category means are equal ( $H_0 : \mu_{\Delta M_{dp}} = \mu_{\Delta M_{wd}} = \mu_{\Delta M_{div}}$)
Test statistic:

$$F = \frac{MS_B}{MS_W}$$

where the mean squares are defined as:

$$MS_B = \frac{SS_B}{k-1}, \qquad MS_W = \frac{SS_W}{\sum n_{\text{adj}} - k}$$

$SS_B$ is the between-group sum of squares, $SS_W$ is the within-group sum of squares, and $k$ is the number of groups (K=3 for our case). The corresponding degrees of freedom are:

$$df_{\text{between}} = k - 1, \qquad df_{\text{within}} = \sum n_{\text{adj}} - k$$

The p-value for the test is given by:

$$p = 1 - F_{df_{\text{between}}, \, df_{\text{within}}}(F)$$

where $F_{df_{\text{between}}, \, df_{\text{within}}}(\cdot)$ is the cumulative distribution function of the F-distribution.

### K.2 Adjusted Two-sample t-Test

Null hypothesis: $\Delta M$ for two category's population means are equal ( $H_0 : \mu_{\Delta M_i} = \mu_{\Delta M_j}$, where $i, j \in dp, wd, div$) Test statistic:

$$t = \frac{\bar{x}_1 - \bar{x}_2}{\sqrt{\frac{s_1^2}{n_{1,\text{adj}}} + \frac{s_2^2}{n_{2,\text{adj}}}}}$$

with degrees of freedom given by the Welch approximation:

$$df \approx \frac{\left( \frac{s_1^2}{n_{1,\text{adj}}} + \frac{s_2^2}{n_{2,\text{adj}}} \right)^2}{\frac{\left( \frac{s_1^2}{n_{1,\text{adj}}} \right)^2}{n_{1,\text{adj}}-1} + \frac{\left( \frac{s_2^2}{n_{2,\text{adj}}} \right)^2}{n_{2,\text{adj}}-1}}$$

and the two-sided p-value is computed as:

$$p = 2 \left( 1 - T_{df}(|t|) \right)$$

where $T_{df}(\cdot)$ is the cumulative distribution function of the t-distribution. Both computations was implemented in `scipy.stats`.

## L  Data Augmentations

We extend the additive data augmentation mode, $\widetilde{X} \approx X + E, E \in \mathbb{R}^{n \times d}$, to a genetic form:

$$\widetilde{X} = f(X, \delta) \approx X + J_f(X)\,\delta,$$

where $f(X, \delta)$ denotes the augmented data produced by transformation parameters $\delta$ (e.g., rotation angle, Gaussian noise ratio), and $J_f(X) = \frac{\partial f(X, \delta)}{\partial \delta}\big|_{\delta=0}$ is the Jacobian of the transformation with respect to these parameters. This linearization corresponds to the first-order term of a Taylor expansion around $\delta = 0$ and assumes that $f$ is differentiable and $\|\delta\|$ is sufficiently small, such that higher-order terms are negligible. Under this general model, $J_f(X)\delta$ represents the local, structured perturbation induced by the augmentation.

- **Gaussian noise:** $f(X,\delta) = X + \delta$. The Jacobian is $J_f(X) = I$ and the perturbation is $J_f(X)\delta = \delta$. This perturbation is data-independent; the same noise $\delta$ is added regardless of the content of $X$.

- **Brightness:** Adjusting global brightness adds the same value to every element of $X$. $f(X,\delta) = X + \delta\mathbf{1}$, where $\mathbf{1}$ is a matrix of all ones having the same shape as $X$. The parameter $\delta$ (the augmentation variable $\delta$) is a scalar controlling brightness increase ($\delta > 0$) or decrease ($\delta < 0$). Since the mapping is linear in $\delta$, the Jacobian is $J_f(X) = \mathbf{1}$ and $J_f(X)\delta = \delta\mathbf{1}$, representing a global additive shift aligned with the data mean.

- **Blur:** Blurring replaces each pixel with a weighted average of its neighbors, written as $f(X,\delta) = X * (k_{id} + \delta_k)$, where $k_{id}$ denotes the identity kernel, a single 1 in the center, and $\delta$ represents a small perturbation of the kernel. Expanding the convolution gives

$$f(X,\delta) = X * (k_{id} + \delta_k) = X + X * \delta_k$$

$J_f(X)\delta = X * \delta_k$ is the result of convolving the image with a small perturbation kernel $\delta_k$, introducing a spatially correlated perturbation.

- **Small rotation:** A tiny rotation by angle $\delta$ is modeled as $f(X,\delta) = XR(\delta)$, where $R(\delta)$ is the rotation matrix. For small $\delta$, $R(\delta) \approx I + \delta A$ with $A$ skew-symmetric (the generator of circular motion). Substituting gives

$$f(X,\delta) \approx X + X(\delta A) = X + J_f(X)\delta, \quad J_f(X) = XA$$

Thus, $J_f(X)\delta = XA\delta$ is a small, data-dependent perturbation moving each point tangentially along the data manifold.

Figure 16 illustrates how Jacobian structures capture different types of augmentations. To further investigate this, we apply the *transform* function from the *torchvision* package in Python to the CIFAR-10 dataset (using the first 200 images) to examine how simple data augmentations affect the trace of the covariance matrix. The results, summarized in Table 9, align with our expectations: except for random cropping, all other augmentations increase the trace of the covariance.

Jacobian structure for different augmentations

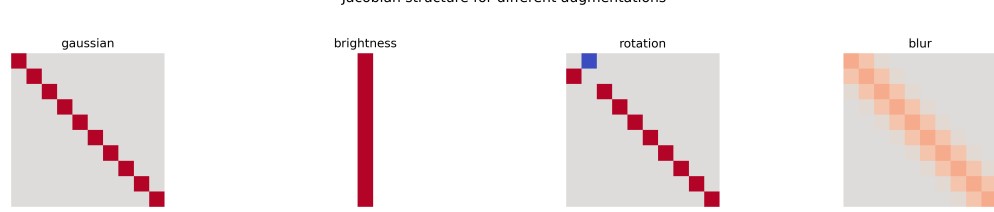

Figure 16: A demonstration for Jacobian structures of different augmentations

Table 9: Trace of covariance matrix (total variance) under different augmentations.

| Augmentation | Trace of Covariance (Total Variance) |
|---|---|
| Brightness (brightness=0.5) | 274.395081 |
| Color Jitter (brightness=0.5, contrast=0.5) | 229.943466 |
| Gaussian Noise (noise ratio=0.1) | 214.028458 |
| Rotation (degrees=30) | 205.135025 |
| Flipping (p=1.0) | 195.942398 |
| Original | 195.942383 |
| Random Crop (size=24) | 175.011536 |

