# OpenReview forum: "Data Diversity as Implicit Regularization: How Does Diversity Shape the Weight Space of Deep Neural Networks?"
_TMLR — Rejected by TMLR_

### Review · Reviewer_bkSB · 2025-09-16

**Summary Of Contributions:**

Based on the empirical observation that increasing data diversity would help neural networks generalize better, the paper investigates how data diversity affects generalization. It first introduces some diversity measures, e.g., Vendi Score, and some random matrix theory. Then it gives an intuitive analysis on the spectral of the weight covariance matrix for least squares regression. Finally, experiments for different data augmentation techniques are conducted and the authors compare these results.

**Additional Comments:**

N.A.

**Audience:**

No

**Audience Explanation:**

The work provides limited new knowledge to understanding neural network trainings.

**Broader Impact Concerns:**

N.A.

**Claims And Evidence:**

No

**Claims Explanation:**

My major concern is about the contribution of this paper. Several points that I am concerning are as follows:

1. While claimed, the authors do not define their metric that quantifies data diversity, at least not explicitly. I do not know what the "weight Vendi Score" is.

2. I appreciate that the authors try to do some theoretical explanation for how data diversity would affect generalization. But as far as my understanding, their analysis has little to do with neural network, needless to say with their experiments. They only did a detailed analysis on least squares regression which is very very different from neural networks, and even for that the analysis only shows the norm of weight matrix is smaller when data is more diversified, which does not explain why generalization is better.

3. The authors seem to be also highlighting their use of random matrix theory. But the work uses very limited amount of it. For example, in Section 3.3, the spectral analysis for least squares regression did not use it. I wonder why this is a contribution/highlight of this paper.

Besides, The authors cite some previous works on the relation between covariance matrix of weights for neural networks and the covariance matrix of the samples, but it is not clear if the assumptions in the previous works are used in the experiments.

**Requested Changes:**

Please revise the paper by considering the weaknesses listed above.

---

> ### Author Response · Authors · 2025-10-05
>
> Thanks for reviewing our manuscript. We provide our responses and clarifications below.
>
> > Q1. about our proposed metric
>
> Thanks for your concern. Our proposed weighted Vendi Score distinguishes between synthetic data augmentation and traditional augmentations, where the original Vendi Score falls short. The motivation comes from the observation that synthetic data
> generated by models often contains broader information beyond the current dataset, whereas traditional augmentations primarily reorganize or transform existing data. The weighted Vendi Score incorporates a scaling factor based on $\rho(X,\tilde X)$, reflecting how much augmented information aligns with the original dataset. Thus, it serves as a complementary metric that captures the distinct informational content provided by synthetic versus traditional augmentations.
>
> > Q2. about regression and generalization
>
> Thank you for the questions. One of our contributions is to characterize the regularization effects of data augmentation by drawing parallels with established techniques like dropout and weight decay, whose mechanisms are well understood. The regression analysis serves more as an illustrative, idealized case, offering intuition rather than a full theoretical validation in DNNs.
>
> It is well known in deep learning theory that smaller weight magnitudes—reflected in lower weight norms—are associated with better generalization. This underlies the effectiveness of techniques of dropout and weight decay, which are widely adopted to reduce overfitting and improve generalization. (We confirm this in our additional norm-matching experiment. Please refer to Section 3.2 and Appendix J.1.) Our analysis shows that diverse data produces similar effects, providing evidence for its role as a regularizer. Given the inherent complexity of deep neural networks, these findings highlight the regularization benefits of data augmentation and extend previous work in this area.
>
> Practitioners have long known that data augmentation improves performance, but not precisely why or how it works. Our work, through spectral analysis and comparative studies with dropout and weight decay, reveals its underlying mechanism.
>
> > Q3. about the use of  Random Matrix Theory
>
> Thanks for your question. Random Matrix Theory (RMT) provides a framework for studying the behavior of eigenvalue spectra in large random matrices. We use RMT, drawing on the Marchenko–Pastur law, heavy-tailed spectra, and the power-law exponent $\alpha$, to interpret weight matrices, complemented by norm- and entropy-based measures that capture both the scale and shape of the spectra, highlighting different aspects of the spectral distribution. The insignificant ANOVA p-value for $\Delta$$\alpha$, when compared with dropout, weight decay, and data augmentation,  indicates that these methods exert similar influences on the shape of the empirical spectral distribution. Additionally, their same directional changes for scale-related metrics relative to the baseline but with different magnitudes highlight the nature of their regularization effects. Thus, our use of RMT extends beyond the power-law exponent $\alpha$ to include a broader set of spectral analysis metrics.
>
> Heavy-tail theory, quantified by the exponent alpha, was introduced by Martin & Mahoney (2021) for analyzing the spectra of deep neural network layers. In DNNs, weight matrices are large (often containing thousands of parameters per layer), and their spectra frequently exhibit heavy-tailed behavior. In contrast, in simpler regression models, the matrices are much smaller, making heavy-tailed RMT analysis with alpha less meaningful. In this section, we instead focus on norms to capture relevant properties of the weights.
>
> > about the assumption in previous literature
>
> We appreciate this thoughtful comment. The non-linear random matrix theory framework introduced by Pennington & Worah (2017) provides an idealized theoretical setting for analyzing how the spectrum of the data covariance matrix propagates through neural network layers. Their validation was primarily conducted in controlled or simulated settings, rather than with large-scale deep networks trained on real-world data.
>
> Our contribution is to extend these theoretical insights into practical scenarios: we empirically analyze the spectral properties of weight matrices from models such as CLIP and BERT trained on diverse datasets. While we do not enforce the exact assumptions of Pennington & Worah (2017), particularly those concerning activation functions that are uncommon in modern DNNs, we find that the qualitative behaviors predicted by their framework (e.g., approximate preservation of the data spectrum) do manifest in practice. We have clarified this distinction in Section 3.3 of the revised paper.
>
> We hope the above responses address all key concerns. Thanks again for your helpful comments and feedback.

---

### Review · Reviewer_p8Kq · 2025-09-17

**Summary Of Contributions:**

**Summary**: This paper explores the role of data diversity in shaping the weight spaces of deep neural network during training, focusing on whether diversity acts as an implicit regularizer.
Specifically, the authors study whether data augmentation have a similar effect as dropout or weight decay.
To this end, they apply Random Matrix Theory to study how data diversity affects the eigenvalue spectra of weight matrices under the assumption that similar specturm of the weight matrices suggests similar generalization.
Comparative experiments among various dataset and tasks are provided to explore the effects of data diversity.

_**Since this reviewer is not familiar with writing codes and conducting experiments, it is hard to assess the experimental part of this paper.
Therefore, the evaluation focuses primarily on the theoretical aspects, conceptual framing, and clarity of the work.**_

**Strengths**
1. This topic is intriguing. Using spectral analysis to study the implicit effects of data diversity is interesting.
2. This paper is well-organized, clear and easy to read for the most part.

**Weaknesses**
1. The theoretical analysis currently feels rather empirical and heuristic, and would benefit from a more rigorous mathematical treatment or clearer formal statements.
2. While the paper’s goal is to study the implicit regularization of deep neural networks, much of the spectral analysis with random matrix theory is illustrated through linear regression examples. Although the authors briefly note that, in the large-data regime, deep network weight spectra may resemble data matrix spectra, this connection could be made more convincing.
3. The paper introduces several technical notions—such as ESD, PL, isotropy, and the decomposition of weight matrices into random noise and signal components—but provides only limited background. For readers not deeply familiar with earlier literature, these sections may be challenging to follow.

4. There are some typos:

Line 4 of page 5: "the covariance matrix" should be the "square of norm"

Line 6 of page 5: "$v_i$" is not defiend.


In the fourth equation of page 5: I am not convincing that the inequality is correct. Consider an extreme case: suppose the pertubation matrix $E = c X$ for some $c\in(0,1)$, then the left hand side $(1-c)^2tr(X^\top X)$ is strictly less than the right hand side $tr(X^\top X)$.

**Audience:**

Yes

**Audience Explanation:**

The topic of data diversity and its implicit regularization effects on deep neural networks is relevant and timely.

**Broader Impact Concerns:**

No major ethical concerns.

**Claims And Evidence:**

Yes

**Claims Explanation:**

The main claims are supported by empirical results and spectral analysis, although some of the theoretical arguments are more heuristic than rigorous.

**Requested Changes:**

1. (**critical**) Add clearer explanations of key concepts (e.g., ESD, PL, isotropy, noise/signal decomposition).
2. (**critical**) Fix typos and improve readability.
3. (**optional**) Strengthen the connection between linear regression analysis and deep neural networks.

---

> ### Author Response · Authors · 2025-10-05
>
> Thank you for taking the time to review our paper and for your thoughtful suggestions. Here are our responses.
>
> **Suggestions in weakness**:
>
> >Line 4 of page 5: "the covariance matrix" should be the "square of norm"
>
> Thanks for pointing it out. We have modified the narratives as suggested.
>
> > Line 6 of page 5: "$v_i$" is not defiend.
>
> Thank you for bringing this up. We have added a clearer explanation about $v_i$ on page 5.
>
> > "In the fourth equation of page 5: I am not convincing that the inequality is correct. Consider an extreme case: suppose the pertubation matrix $E=cX$ for some $c \in (0, 1)$, then the left hand side $(1-c)^2 \operatorname{tr}(X^\top X)$ is strictly less than the right hand side   $\operatorname{tr}(X^\top X)$. "
>
> Thanks for your concern and let us clarify it here: When $E=cX$ and $c \in (0, 1)$, it violates the assumption on page 5: “ Under the assumptions that on the augmentation noise has zero mean and is independent of the data, we have $\mathbb{E}\bigl[X^\top E] = 0$ and $\operatorname{tr}(X^\top E)$ aligns with the original data direction, thus…”. In this extreme case ($E=cX$), the perturbation $E$ is not independent noise but fully collinear with the data $X$, which isn't really "noise" in the way data augmentation is usually understood. In fact, this case is closer to a rescaling of the data rather than noise augmentations.
>
> If we ignore the assumption stated in our paper, algebraically, when $E=cX$, then
>
> $$\sum_i \tilde{\lambda_i} = \operatorname{tr}\big((X+cX)^\top (X+cX)\big) = \operatorname{tr}((1+c)^2 X^\top X)= (1+c)^2 \operatorname{tr}(X^\top X)$$
>
> Thus, if c>0, the inequality still holds.
>
> **Requested Changes**
> > 1. (critical) Add clearer explanations of key concepts (e.g., ESD, PL, isotropy, noise/signal decomposition).
>
> Thanks for your recommendation. We include a brief discussion of these concepts in the Preliminary section and provide a more detailed explanation and demonstration in Appendix D.1.
>
> > 2. (critical) Fix typos and improve readability.
>
> We apologize for the typos and have corrected them.
>
> > 3. (optional) Strengthen the connection between linear regression analysis and deep neural networks.
>
> Thanks for your valuable suggestion. We agree that bridging the theoretical gap between linear models and deep neural networks is an important yet challenging research direction. In our paper, the linear regression analysis is intended as an illustrative case to build intuition. It demonstrates that increasing data variance via augmentation inflates the eigenvalues of the data correlation matrix, thereby reducing the norm of the optimal weights. This mechanism aligns with a well-established principle in deep learning: smaller weight magnitudes are associated with better generalization and are the goal of explicit regularizers like weight decay.
>
> While a complete theoretical derivation for deep architectures lies beyond the present scope, our extensive empirical results substantiate this intuition: data augmentation induces spectral changes analogous to those from dropout and weight decay across complex deep models. We have added text to Section 3.3 to clarify this framing and better contextualize the linear analysis as a source of theoretical motivation for our empirical study. Developing a formal treatment for deep networks remains a compelling direction for future work.
>
> Thanks again for your effort in reviewing our paper and insightful recommendations. All revisions are highlighted in blue in the revised paper. Hope our responses address your concerns.

---

### Review · Reviewer_RHTB · 2025-09-22

**Summary Of Contributions:**

This paper studies how training data diversity acts as an implicit regularizer for deep neural networks. The authors use Random Matrix Theory (RMT) to analyze the spectral properties of weight matrices under three regimes, i.e., data augmentation, dropout, and weight decay, and observe that diversifying inputs yields spectral changes similar to explicit regularization, and most closely resembles dropout. They further introduce a Weighted Vendi Score that combines diversity and alignment to the original data in embedding space, and use it to explain when traditional vs. synthetic augmentations help. Experiments span vision (CLIP ViT-B/32, ResNet-50 fine-tuning; ResNet-18 from scratch) and language (BERT fine-tuning), with statistical testing across layers and datasets. Key claims and contributions are stated clearly in the abstract and introduction and reiterated in a bullet list.

- Clear conceptual framing and use of RMT to connect input diversity → weight spectra → generalization, with heavy-tailed/self-regularization context.

- Consistent empirical evidence that augmentation ≈ dropout in spectral behavior across models/datasets, backed by per-layer analyses and statistical tests.

- Practical evaluations include ID/OOD accuracy and calibration (ECE), not just spectra.

**Audience:**

Yes

**Audience Explanation:**

The work addresses a foundational question, i.e., how and why data diversity helps, using RMT, which many TMLR readers care about. Moreover, It provides diagnostic tools and actionable insights for augmentation design and synthetic data mixing, directly relevant to practice.

**Claims And Evidence:**

Yes

**Claims Explanation:**

The main claims are supported by empirical evidence and spectral analysis.

**Requested Changes:**

Clarify correlation vs. causation: Temper causal language or add controlled interventions: e.g., match Frobenius/spectral norms across regimes and test whether residual differences persist. Include random-label or frozen-feature controls to isolate regularization vs. representation effects.

---

> ### Author Response · Authors · 2025-10-05
>
> Thank you for your insightful comments. We conducted additional experiments as suggested. Here are our responses.
>
> **Norm matching experiments**: the baseline model is fine-tuned without any techniques such as dropout, weight decay, or data augmentations on the CIFAR-10 dataset.  We then employ fine-tuned models obtained through any of the three methods described in the paper as our target models. The norm-matching approach we applied is described below:
>
> Let the baseline model weights be $W_{\text{base}}$ with Frobenius norm $|| W_{base} ||$, and the target model weights be $W_{target}$ with norm $|| W_{target}||$. The scaling factor is defined as $ s = \frac{|| W_{target}||}{W_{base}||}$ and the scaled weights are given by  $W_{base-scaled} = s \cdot W_{base}$. This ensures that the baseline model is matched in terms of norm with three different approaches.  Finally, we use matched or weight-rescaled baseline models to predict on the test set.
>
> The results (Appendix J.1) demonstrate that weight decay–based norm matching causes the most substantial accuracy changes, while dropout and data augmentation lead to only minor deviations from the baseline. These findings are consistent with the greater magnitude of weight norm changes induced by weight decay compared to the smaller effects of dropout and data augmentation (refer to Figures 3 and 4 in our paper). Thus, this experiment suggests that larger differences in Frobenius norm significantly affect model prediction performance, confirming the importance of weight norm in model generalization and highlighting the regularization effect provided by data diversity.
>
> **Frozen Image Encoders:**  In this experiment, we freeze the image encoder in CLIP, and only the weights in the last classification layer are allowed to be updated.
>
> Result (Appendix J.2): We observe the same directional changes in both alpha and the Frobenius norm across all three approaches. Moreover, the directional shifts in alpha, Frobenius norm, and spectral norm are closely aligned for dropout and data augmentations, suggesting that these methods influence the weights in a similar manner.
>
> **Random Labels:** We randomly shuffle 30% and 50% of labels in the training set of CIFAR-10. The results (Appendix J.3) consistently show that all three methods (weight decay, dropout, and data augmentation) reduce both the Frobenius norm and spectral norm, with the relative magnitude of Frobenius norm shrinkage on average following the order: weight decay > data augmentation > dropout.
>
> At the 0.3 corruption level, scale-related metrics such as alpha and entropy exhibit slightly different directional changes across methods, but the adjusted ANOVA test confirms that these differences in alpha and entropy are not statistically significant, aligning with prior observations in the paper. At the 0.5 corruption level, a similar pattern emerges: all three methods reduce Frobenius norm, spectral norm, and alpha, while entropy again diverges in directionality: this time with dropout aligning more closely with data augmentation, but differing from weight decay. These differences in entropy are also statistically non-significant according to the adjusted ANOVA test, consistent with reported findings in the paper.
>
> Overall, the frozen image encoders and random labels experiments suggest that the regularization effect of data diversity introduced through augmentation is independent of feature learning and behaves in a manner similar to dropout, particularly in terms of the magnitude of norm decrease, whereas weight decay exerts a stronger influence. This confirms previous findings, and its regularization is different from representation effects. We added one paragraph for a simple description of these additional experiments in Section 4.2.1 and provided detailed results and tables in Appendix J.
>
> Lastly, we have also refined overly casual language in the paper. All revisions are highlighted in blue. Thanks again for helping us to improve this work.

---

### Comment · Action_Editor_CgR5 · 2025-10-29
**Questions**

Dear Authors and Reviewers,

Thank you for the reviews and discussions. While reading the comments and the paper again, I got some questions. So let me ask the authors before we make the final recommendation:

1.
  > Data augmentation can be seen as adding a perturbation matrix E, so augmented data $X = X + E$, $E \in \mathbb{R}^{n\times d}$

  How does $E$ represent non-additive transforms such as cropping and rotation?

2.
  > $\operatorname{tr}\left(X^{\top} X\right)+2 \operatorname{tr}\left(X^{\top} E\right)+\operatorname{tr}\left(E^{\top} E\right) >\operatorname{tr}\left(X^{\top} X\right)$

  This might not be true, for instance, if $E = 0$. Also, is this inequality meant to hold with probability 1?

3.
  > Treating the distribution of $\Delta M$ across layers for each category as time series data

  Could the authors elaborate on what time series data we obtain here? Also, what is a category here?

4.
  > $tr(X^{\top} E)$ aligns with the original data direction

  What does this mean mathematically?

5.
  > $(\mathbb{E}[f'(z)])^2 = 0$

  The constant function $f: z \mapsto 0$ seems to satisfy this condition, but then, $(Y^{(l)})^\top Y^{(l)} = 0$ (if $Y^{(l)}$ represents the output of a layer with activation $f$), so the equation displayed below that part does not hold unless $X = 0$. Could you provide a formal statement and a proof?

6.
  > we apply the adjusted ANOVA (F-test) and autocorrelation-aware pairwise t-tests

  > All of these observations are statistically confirmed by adjusted ANOVA and t-test

  What null hypotheses do they test? With what statistics?

7. In the experiments, different types of architectures are used, and even each of them has different types of layers of different sizes. For example, a ResNet has several convolutional layers followed by a linear layer at the end. Are the metrics the authors use not sensitive to these types or the size of the layers?

8. In the definition of the weighted Vendi Score, how is $\tilde{X}$ defined exactly? It needs to have the same sample size as that of $X$?

9.
  > embedding-based Vendi Score

  What embedding model was used?

10. In each scatter plot, what are the different dots?

---

> ### Author Response · Authors · 2025-11-06
>
> Dear Editor and Reviewers,
>
> Thank you for reviewing my paper and insightful questions. Here are our responses.
>
> > Reply to Q1:
>
> The formulation $\widetilde{X}=X + E$ is an abstract representation of data augmentation. It can represent rotation but not cropping.  In the revised manuscript, we extend the additive data augmentation mode to a genetic form in Appendix L:  $\widetilde{X} = f(X, \delta) \approx X + J_f(X)\,\delta$, where $f(X,\delta)$ denotes the augmented data produced by transformation parameters $\delta$ (e.g., rotation angle) and  $J_f(X)=\frac{\partial f(X,\delta)}{\partial \delta}\big|_{\delta=0}$ is the Jacobian of the transformation with respect to these parameters. This linearization corresponds to the first-order term of a Taylor expansion around $\delta=0$ and assumes that $f$ is differentiable and $\|\delta\|$ is sufficiently small, such that higher-order terms are negligible.
>
> A small rotation by angle $\delta$ is modeled as $f(X,\delta)=X R(\delta)$, where $R(\delta)$ is the rotation matrix. For small $\delta$, using Taylor expansion, $R(\delta)\approx I+\delta A$ with $A$ skew-symmetric. For non-additive transformations such as cropping, we represent the augmentation as a linear mapping $\widetilde{X} \approx X P$, so that $\sum_i \tilde{\lambda}_i = tr( X^\top X PP^\top)$. Here, $PP^\top\preceq I$ (e.g., cropping) decreases total variance, while $PP^\top \succeq I$(e.g., color jitter) increases it. We empirically validated these behaviors on CIFAR-10 (see Table 9).
>
> > Reply to Q2:
>
> We have revised this derivation and now distinguish two important cases based on the relationship between E and X:
> 1. When augmentation noise has zero mean and is uncorrelated with the data, $\mathbb{E}[X^\top E]=0$ and $\mathbb{E}[E^\top E]>0$. This means that additive, zero-mean perturbations increase total variance on average.
> 2. When augmentations depend on $X$, such as blur or small rotations, the cross term $tr(X^\top E)$ can be positive or negative. If $E$ perturbs samples along the existing feature directions (the goal of data augmentations without distribution shift), then $tr(X^\top E)>0$, leading to $\sum_i \tilde{\lambda_{i}} > \sum_i \lambda_{i}$.
>
> > Reply to Q3:
>
> Take Figure 3 as an example, where the data shown represent $\Delta_M$. $M \in$ {$\alpha$, entropy, Frobenius norm, and spectral norm}. Each corresponds to a separate plot (indicated by the y-axis). Let’s focus on the first plot. This plot shows $\Delta_\alpha$, where each data point corresponds to the change in $\alpha$ under dropout, weight decay, or data augmentation compared to the baseline. This is why each point is scattered around aero.
>
> For this plot, the data can be represented as a time series in the following form:
> | Layer | Δα_dropout_avg | Δα_weight_decay_avg | Δα_data_augmentation_avg |
> |:------|:----------------|:--------------------|:--------------------------|
> | Lay1  | xx              | xx                  | xx                        |
> | Lay2  | xx              | xx                  | xx                        |
> |... | ...              | ...                  | ...                        |
>
>  > Reply to Q4:
>
> In the revised version, we have restructured this part to avoid the ambiguous term “aligns with the data direction” and instead explain the role of the cross term $tr(X^\top E)$ directly. Mathematically, $tr(X^\top E)=\sum x_i^{\top} e_i = \sum \langle x_i, e_i \rangle$, which is the sum of inner products between each original sample $x_i$​ and its augmentation perturbation $e_i$.
> - If $tr(X^\top E) >0 $, then, on average, the perturbations $e_i$​ in similar directions as the original data $x_i$.
> - If $tr(X^\top E) \approx 0$, perturbations are uncorrelated to the data, as in zero-mean Gaussian noise.
> - If $tr(X^\top E) <0 $, augmentations oppose the original structure, potentially reducing total variance.
>
> In the revised manuscript, we explicitly discuss how the sign of $tr(X^\top E)$ determines whether augmentation amplifies or reduces the total variance. So, the earlier “alignment” statement corresponds mathematically to the case $tr(X^\top E)$>0, meaning the perturbations add energy along the same principal directions of X.

---

> ### Author Response · Authors · 2025-11-06
>
> >  Reply to Q5:
>
> Our statement directly summarizes the result of Pennington & Worah (2017), who analytically studied the spectrum of the random Gram matrix $M = \frac{1}{m} Y Y^{\top}, Y = f(WX)$
> where $X \in \mathbb{R}^{n_0 \times m}$ (input data) and $W \in \mathbb{R}^{n_1 \times n_0}$ (random weights with i.i.d. Gaussian entries). The Gram matrix $M \in \mathbb{R}^{n_1 \times n_1}$ has a shape parameter $c = \frac{m}{n_1}$,
> the same ratio that controls the Marchenko-Pastur (MP) law governing the spectrum of large random covariance matrices.
>
> They introduced two scalars that capture the effect of the nonlinearity: $\eta = \mathbb{E}\left[ f(aZ)^2 \right], \zeta = \left( a \mathbb{E}\left[ f'(aZ) \right] \right)^2, Z \sim \mathcal{N}(0,1), a = \sigma_w \sigma_x.$
>
> By expanding the moments of $M$ through a combinatorial enumeration of non-crossing (outer-planar) graphs and resumming them into a generating function,  Pennington & Worah (2017) showed that the Stieltjes transform $G(z)$ of the limiting eigenvalue distribution satisfies a self-consistent quadratic equation. When $\zeta = 0 $ ( that is, when$(\mathbb{E}[f'(z)])^2 = 0$), this equation simplifies exactly to the MP form: $ zG^2 + ((1-c)z-1)G + c = 0$, which implies that the nonlinear layer’s spectrum has the same MP form as the input covariance matrix $\frac{1}{m}X X^{\top}$. In this regime, the nonlinearity is said to be isospectral, meaning it does not distort the eigenvalue distribution as signals propagate through layers. The spectrum of $\frac{1}{m}YY^T$ becomes identical to the spectrum of $\frac{1}{m}XX^T$ if and only if the layer widths are matched ($n_0 = n_1$).
>
> The constant function $f(z)=c$ satisfies $(\mathbb{E}[f'(z)])^2=0$, but it leads to a trivial output spectrum. This function is excluded by the assumptions of Pennington & Worah's (2017) analysis.
>
> >  Reply to Q6:
>
> For both tests, we recompute the effective sample size $n_{adj}$ by lag-1 autocorrelation $\rho$. $n_{adj} = n*\frac{1-\rho}{1+\rho}$
>
> **Adjusted ANOVA Test:**
>
> Null hypothesis: $\Delta M $ for all three different category means are equal ( $H_0: \mu_{\Delta M_{dp}} = \mu_{\Delta M_{wd}} = \mu_{\Delta M_{div}} $).
>
> Test statistic: $F = \frac{MS_B}{MS_W}$, where the mean squares are defined as:
> $MS_B = \frac{SS_B}{k - 1}, MS_W = \frac{SS_W}{\sum n_{adj} - k}$. $df_{\text{between}} = k - 1, df_{\text{within}} = \sum n_{adj} - k$.
>
> **Adjusted Two-sample t-Test:**
>
> Null hypothesis: $\Delta M $ for two category's population means are equal ( $H_0: \mu_{\Delta M_{i}} = \mu_{\Delta M_{j}}$, where $i, j \in {dp, wd, div}$)
>
> Test statistic: $t = \frac{\bar{x}_1 - \bar{x}_2}{denominator}$,
>
> where $ \text{denominator} = \sqrt{\frac{s_1^2}{n_{1,adj}} + \frac{s_2^2}{n_{2,adj}}}$
>
> with degrees of freedom given by the Welch approximation:
> $df \approx
> \frac{\left( \frac{s_1^2}{n_{1,adj}} + \frac{s_2^2}{n_{2,adj}}\right)^2}
> {\frac{ \left( \frac{s_1^2}{n_{1,adj}} \right)^2 }{n_{1,adj} - 1} +
>  \frac{ \left( \frac{s_2^2}{n_{2,adj}} \right)^2 }{n_{2,adj} - 1}}$
>
> >  Reply to Q7:
>
> The weight matrices we extract and analyze are the main structural components of the models, excluding normalization layers such as BatchNorm and LayerNorm. For example, for the CLIP ResNet-50 model, we include the convolutional layers, each of the ResNet blocks (1–4), and the final classification layer. For the ViT model, we analyze the attention output projection and the two MLP layers within each transformer block.
>
> For scale-dependent metrics such as the Frobenius norm and spectral norm, we observe distinct patterns in both magnitude and directional changes between the final classification layer and the previous layers. Additionally, the magnitude of these changes differs between the shallow and deep layers.
>
> For shape-related metrics such as the power-law exponent ($\alpha$) and matrix entropy, we observe some directional differences between the final classification layer and the other layers: both ResNet-50 and ViT-B/32 exhibit similar trends in $\alpha$ (see Figures 3 and 9). However, changes in matrix entropy are less pronounced compared to the norm-based metrics. We discussed these observations in the second paragraph of Section 4.2.1 and the first paragraph on page 9.
>
> >  Reply to Q8:
>
> $X$ is the oprginal data and $\widetilde{X}$ is defined as the augmented data. Yes, $\widetilde{X}$ is required to be the same size as $X$. In section 4.3, $D_{syn}$ and $D_{vf}$ are constructed to match the size of the original dataset $D$. For $D_{add}$, since synthetic samples increase the dataset size, we randomly sample real data from $D$ so that the combined size of the synthetic data and real data equals that of the original dataset.

---

> > ### Author Response · Authors · 2025-11-06
> >
> > >  Reply to Q9:
> >
> > Our embedding model is based on pretrained models such as CLIP or BERT, which provide embeddings used to preprocess image or text data. Since we experiment with different CLIP backbones (ResNet and ViT), the embedding model aligns with the chosen backbone structure. For instance, if CLIP uses a ResNet-50 backbone, the embedding model is ResNet; otherwise, it is ViT for vision-related tasks.
> >
> > >  Reply to Q10:
> >
> > Figures 3 and 4: These are discussed in Question 3.
> >
> > Figure 5: Each dot represents one of the ten different data augmentation techniques, corresponding to those shown in Figure 2. The x-axis indicates the diversity score (Vendi Score) of the augmented data, while the y-axis shows the test accuracy of models fine-tuned using the corresponding augmented datasets.
> >
> > Figure 6: For synthetic data augmentations, each dot represents a different mixing ratio between real and synthetic data, with ratios [0.1,0.3,0.5,0.7,0.9,1.0]. The size of each dot reflects the proportion of synthetic data used. For example, the largest dot (1.0) represents fully synthetic data, while the smallest dot (0.1) represents 10% synthetic data. Purple dots, in contrast, represent traditional data augmentation methods (EDA), all shown with equal dot sizes. The x-axis shows the proposed weighted VS values for each dataset, and the y-axis shows the test accuracy of models fine-tuned on those augmented datasets. We have added a more detailed description in Figures 5 and 6.
> >
> > Finally, we sincerely appreciate the efforts of the AE and reviewers, and we hope our revisions address all your concerns. We would be glad to provide further clarifications if needed.

---

### Decision · Action_Editor_CgR5 · 2025-11-15

**Recommendation:** Reject

**Additional Comments:**

The recommendations from the reviewers were Leaning Accept, Leaning Accept, and Reject. After carefully reading the paper and the comments, I do not recommend accepting this submission this time.

The main reason of the rejection is the clarity and the accuracy of the claims. Reviewers find the topic of this work interesting. The authors addressed many of the concerns raised in the discussion and suggested great modifications to the paper, but I believe the work deserves a revision and a resubmission for making the claims and presentation accurate. I encourage the authors to resubmit a revised paper to TMLR or another venue.

**Audience:**

Yes

**Audience Explanation:**

Two reviewers expressed that the work should interest the TMLR audience:
- "The topic of data diversity and its implicit regularization effects on deep neural networks is relevant and timely"
- "The work addresses a foundational question"
- "provides diagnostic tools and actionable insights for augmentation design and synthetic data mixing, directly relevant to practice"

One reviewer is not convinced that the work has made the relevant contributions, despite the claims of the paper.

**Claims And Evidence:**

No

**Claims Explanation:**

This paper studies data augmentation in training deep neural networks, to investigate the hypothesis that data augmentation has an implicit regularization effect. The paper demonstrates the importance of the spectral distribution of the Gram matrix, and how it can be affected by data augmentation. It also discusses how the random matrix theory of Pennington and Worah (2017) can be related to the relationship between the spectra of the input matrix and the next layer of the network. The authors use the Vendi Score to measure data diversity increased by data augmentation. The experiments reveal correlation between accuracy and the Vendi Score, some parameter of the eigenvalue distribution, the matrix entropy, the Frobenius norm, and the spectral norm of the weights. The authors propose a weighted version of the Vendi Score which is capable of distinguishing simple data augmentation from data synthesized by generative models.

There are some concerns about the theory part raised by the reviewers and myself:
- The theoretical claims seem to lack formality and accuracy.
- The theory studies only least squares regression, and there is a gap from neural networks and their experimental setups.
- Little contribution to the claimed connection between the data augmentation and the random matrix theory.
- There seems to be a gap between the claim made by this paper and the cited result of Pennington and Worah (2017). Their theory studies the asymptotic behavior of the spectrum as both the dimension and the sample size go to infinity. They also mention that the theory is not applicable to multiple layers.
- The types of data augmentation considered in the theory seem quite limited compared to what the introduction may imply and those used in the experiments (although the revision of the main text and Appendix L the authors updated during the discussion does improve the clarity).

The claims about the empirical results are supported by the experiments. There were some clarity issues, which the authors addressed during the discussion. However, the message from the experiment using the weighted Vendi Score is less clear because the sign of the correlation depends on the data augmentation we use. A reviewer expressed that the presentation of the experiments might lack some details.

**Resubmission Of Major Revision:**

The authors may consider submitting a major revision at a later time.